# Visceral obesity and insulin resistance associate with CD36 deletion in lymphatic endothelial cells

Vincenza Cifarelli [1✉], Sila Appak-Baskoy [2,3], Vivek S. Peche[1], Andrew Kluzak[1], Trevor Shew[1], Ramkumar Narendran[1], Kathryn M. Pietka[1], Marina Cella [4], Curtis W. Walls[1], Rafael Czepielewski [4], Stoyan Ivanov[4], Gwendalyn J. Randolph [4], Hellmut G. Augustin [2,3] & Nada A. Abumrad [1,5✉]

Disruption of lymphatic lipid transport is linked to obesity and type 2 diabetes (T2D), but regulation of lymphatic vessel function and its link to disease remain unclear. Here we show that intestinal lymphatic endothelial cells (LECs) have an increasing CD36 expression from lymphatic capillaries (lacteals) to collecting vessels, and that LEC CD36 regulates lymphatic integrity and optimizes lipid transport. Inducible deletion of CD36 in LECs in adult mice (Cd36$^{\Delta LEC}$) increases discontinuity of LEC VE-cadherin junctions in lacteals and collecting vessels. Cd36$^{\Delta LEC}$ mice display slower transport of absorbed lipid, more permeable mesenteric lymphatics, accumulation of inflamed visceral fat and impaired glucose disposal. CD36 silencing in cultured LECs suppresses cell respiration, reduces VEGF-C-mediated VEGFR2/ AKT phosphorylation and destabilizes VE-cadherin junctions. Thus, LEC CD36 optimizes lymphatic junctions and integrity of lymphatic lipid transport, and its loss in mice causes lymph leakage, visceral adiposity and glucose intolerance, phenotypes that increase risk of T2D.

[1] Center for Human Nutrition, Department of Medicine, Washington University School of Medicine, St. Louis, USA. [2] European Center for Angioscience, Medical Faculty Mannheim, Heidelberg University, Heidelberg, Germany. [3] Division of Vascular Oncology and Metastasis, German Cancer Research Center (DKFZ-ZMBH Alliance), Heidelberg, Germany. [4] Department of Pathology and Immunology, Washington University School of Medicine, St. Louis, USA. [5] Department of Cell Biology and Physiology, Washington University School of Medicine, St. Louis, USA. ✉email: cifarelli@wustl.edu; nabumrad@wustl.edu

Dietary lipids are transported to the circulation predominantly as chylomicrons via the intestinal lymphatic system, which drains into the subclavian vein[1]. Chylomicrons enter the intestinal lymphatic capillaries, or lacteals, through the open "button-like" vascular endothelial (VE)-cadherin junctions localized between adjoining lymphatic endothelial cells (LECs)[2]. The "zippering" of VE-cadherin junctions, through deletion of neuropilin1 and vascular endothelial growth factor receptor 1 (VEGFR1), prevents chylomicron uptake by lacteals and fat absorption protecting against diet-induced obesity and systemic glucose intolerance[2]. Similar metabolic protection is achieved by deletion of vascular endothelial growth factor (VEGF)-C[3], or of VEGF-C regulated Notch ligand delta like−4 (DLL4)[4], both important for lacteal maintenance and remodeling[5]. On the other hand, mice heterozygous for the transcription factor prospero-related homeobox 1 (Prox1) important for LEC lineage commitment[6] show defective, leaky lymphatic vessels and adult-onset obesity[7]. Disruption of lymphatic vessel integrity[8,9] and attenuation of VEGF-C induced lymphangiogenesis[10] have been reported as pathophysiological phenotypes in mice models of obesity and type 2 diabetes (T2D). In humans, lymphatic dysfunction is associated with obesity, diabetes, and aging, which increases the risk of cardiovascular disease, lymphedema and age-related neurological decline[11–15], but our knowledge of the mechanisms underlying lymphatic dysfunction and its link to disease remains limited.

The fatty acid transporter CD36/FAT is a transmembrane scavenger receptor widely expressed in tissues[16] and cell types including endothelial cells[17,18]. CD36 recognizes long-chain fatty acids (FAs)[16,19], lipoproteins[20], pathogen-associated lipids[21], in addition to non-lipid ligands[22,23]. Studies in rodents[24,25] and people[26,27] have shown that CD36 facilitates tissue uptake of non-esterified FAs, and that of FA released from very low-density lipoproteins (VLDL)[28]. CD36 deficient ($Cd36^{-/-}$) mice have reduced FA uptake by peripheral tissues, and this reduction was recapitulated in mice with specific deletion of endothelial cell $Cd36$, highlighting its regulatory role in tissue FA uptake[18]. There is little information on CD36 expression level or its role in the lymphatic endothelium. Although human dermal LECs were reported to express CD36 mRNA[29], cutaneous lymphatic vessels were found to have low CD36 protein content[30] and there is no evidence for CD36 expression in LECs of the small intestine. We had previously shown that $Cd36^{-/-}$ mice have reduced lipid secretion into the cannulated mesenteric lymph duct following duodenal lipid infusion, and at the time attributed this impairment to the defective generation of chylomicrons by enterocytes devoid of CD36[31,32].

In this study, we show that CD36 is highly expressed in intestinal lymphatics (lacteals and submucosa) and that $Cd36^{-/-}$ mice have discontinuous VE-cadherin junctions in gut mucosa (lacteals) and submucosa lymphatic vessels. To understand the role of CD36 in LEC function, we generated a mouse with inducible $Cd36$ deletion in LECs (Prox1-CreER$^{T2}$-tdTomato$Cd36^{-/-}$, hereafter referred to as $Cd36^{\Delta LEC}$) and assessed VE-cadherin morphological status in gut lymphatics, lymph transport, and metabolic phenotypes. Mechanistic studies with cultured LECs highlighted the importance of CD36 in the regulation of oxidative and glycolytic metabolism. Here, we show that inducible LEC CD36 deletion in adult mice causes leaky lymphatic vessels in the mesenteric region, accumulation of inflamed visceral adipose tissue, and spontaneous late-onset obesity. CD36 silencing in LECs inhibits fatty acid oxidation (FAO) and increases glycolytic rates. The switch in metabolic fuel associates with reduced VEGF-C signaling to VEGFR2/AKT, which impairs LEC migration, tube formation and monolayer integrity.

## Results

**$Cd36^{-/-}$ mice have shorter lacteals and disrupted LEC VE-cadherin junctions.** CD36 is abundantly expressed in blood microvascular vessels[18] but its expression in gut lymphatics is unstudied. Immunohistochemical analysis of lacteals in the jejunum of C57BL/6 mice (hereafter referred to as wild type, WT) showed CD36 staining (green fluorescence) colocalizing with the lymphatic vessel marker endothelial hyaluronan receptor 1 (LYVE−1, red fluorescence) (Fig. 1a, left panel, individual staining shown in Supplementary Fig. 1). Similarly, CD36 expression and colocalization with LYVE-1 were observed in collecting lymphatic vessels in the submucosa (Fig. 1a, right panel and individual staining in Supplementary Fig. 1). Not all LYVE-1$^+$ cells express CD36, which is in line with gene expression heterogeneity described in LECs within lymphatic vessels[33]. Flow-cytometric analysis of LECs (CD45$^-$ CD90.2$^+$ CD31$^+$ cells) (Fig. 1b) isolated from the jejunal mucosa (lacteals) and submucosa, and from the collecting vessels in the mesentery of WT mice (Fig. 1c) showed increasing CD36 expression from lacteals (mean fluorescence intensity, MFI: 2952) to submucosa (MFI: 4027) to mesenteric collecting vessels (MFI: 9010). CD36 staining was absent in $Cd36^{-/-}$ gut LECs and following IgG isotype control antibody staining used at the same concentration of the CD36 antibody (Fig. 1c).

To examine if lacteal structure is altered by CD36 deletion, we performed immunostaining of LYVE-1 and alpha-smooth muscle actin (αSMA) in intestine whole-mounts. $Cd36^{-/-}$ lacteals in the jejunum were slightly shorter as compared with those of WT mice ($5.2 \pm 0.9 \times 10^2$ μm vs. $7.5 \pm 0.3 \times 10^2$ μm; $P < 0.01$) often not extending to the tip of villi (Fig. 2a, b). The density of smooth muscle fibers surrounding the lacteals and the length of villi appeared similar in the two mice groups (Fig. 2a, b). Gene expression of $Vegfc$ and $Dll4$, critical to lacteal proliferation and maintenance[3,4] was similar (Fig. 2c). Lymphatic function is regulated by the junctions between adjoining LECs, which consist of assembled multiprotein complexes that include VE-cadherin[2,34]. We examined the morphology of LEC VE-cadherin junctions in WT and $Cd36^{-/-}$ whole-mount intestines. Lacteals in WT mice presented both open and closed LEC junctions (Fig. 2d) as reported in previous studies[2,4,34,35]. In contrast, lacteals from $Cd36^{-/-}$ mice showed fragmented VE-cadherin staining (Fig. 2d). Lymphatic vessels in the submucosa of WT mice had continuous VE-cadherin junctions, and by comparison the junctions in $Cd36^{-/-}$ mice were more discontinuous and fragmented (Fig. 2d). We next examined by microscopy lymph transport following an intragastric load of the fluorescently labeled tracer long-chain fatty acid BODIPY C16[7]. BODIPY fluorescence rapidly increased in the circulation of $Cd36^{-/-}$ mice (Supplementary Fig. 1b) recapitulating the rapid blood appearance of absorbed TGs in $Cd36^{-/-}$ mice[31], and reflecting compromised integrity of blood endothelial vessels, as shown with Tie2-driven CD36 deletion[17]. Immunohistochemical analysis of whole-mount mesentery showed no differences in expression of FoxC2, which identifies lymphatic valves and alpha-smooth muscle actin, α-SMA between $Cd36^{-/-}$ and WT mice (Supplementary Fig. 2). These findings suggested that CD36 deletion disrupts the organization of VE-cadherin at LEC junctions in lacteals and collecting vessels and that chylomicrons reach the circulation directly and independently of lymphatic transport in $Cd36^{-/-}$ mice.

**CD36 deletion in LECs leads to obesity and leaky gut lymphatics.** To help dissect out the role of CD36 deletion in the regulation of lymphatic endothelial VE-cadherin junctions, we crossed $Cd36^{fl/fl}$[17,18] mice with Prox1-CreER$^{T2}$-tdTomato mice[36] to generate a mouse with $Cd36$ deletion in LECs ($Cd36^{\Delta LEC}$). The deletion was induced by intragastric administration of tamoxifen at 8 weeks

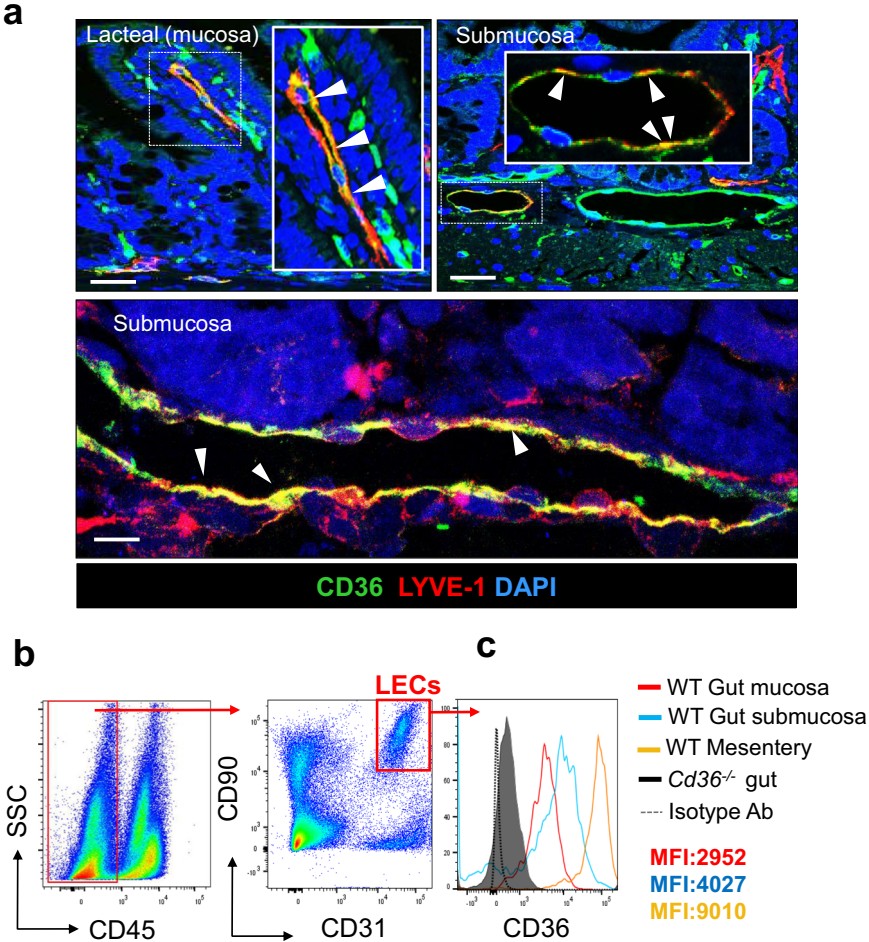

**Fig. 1 Cd36 expression in gut and mesenteric lymphatic vessels. a** Representative staining for CD36 (green) and its colocalization with lymphatic vessel endothelial hyaluronan receptor 1, LYVE-1, (red) in lymphatic capillaries (lacteals) and in collecting vessels within the submucosa in C57BL/6 wild-type (WT) mice. Nuclei stained with DAPI (blue). Images were acquired using a Zeiss LSM 880 Airyscan Confocal microscope, ×20 and ×40 objective. Scale bar: 50 μm (top panel) and 10 μm (bottom and panel). **b** Flow cytometry to identify intestinal and mesenteric lymphatic endothelial cells (LECs) (CD45⁻ CD90.2⁺CD31⁺). **c** CD36 expression in LECs isolated from gut mucosa (red) and submucosa (blue), and the mesentery (orange) in WT mice. Jejunum from $Cd36^{-/-}$ mice and IgG isotype control antibody used at the same concentration as CD36 antibody were included to document CD36 antibody specificity. Mice ($n = 6$) used in (**a–c**) are 12-week old. SSC side scatter, MFI mean fluorescence intensity, CD cluster of differentiation. Data are representative of three experiments with $n$ representing the number of mice per group.

of age, every other day for two weeks, to activate Cre recombinase, which also drives tdTomato reporter expression. Tamoxifen was similarly administered to flox negative, Cre positive littermate mice referred to as controls. Validation of the $Cd36^{\Delta LEC}$ mouse model and metabolic studies were conducted at one week after the last tamoxifen treatment (11-week old) and at 10–11 weeks later (~20-week old) with mice maintained on a standard chow diet (Supplementary Fig. 3a). Flow cytometry analysis confirmed that tdTomato⁺ cells were primarily (~95%) LECs (CD45⁻CD90.2⁺ CD31⁺) and that tamoxifen treatment reduced CD36 expression in tdTomato⁺ cells from 63 to 15% (Supplementary Fig. 3b). Whole-mount staining showed that tdTomato (red) associated with LYVE-1⁺ (green) which identified lacteals (Supplementary Fig. 3c). CD36 deletion in LECs was further confirmed by qRT-PCR in sorted intestinal tdTomato⁺ LECs from $Cd36^{\Delta LEC}$ mice while $Prox1$ mRNA was not altered (Supplementary Fig. 3d).

The $Cd36^{\Delta LEC}$ males and female mice developed spontaneous obesity with age (20-week old) while on the standard chow diet; body weight was higher in $Cd36^{\Delta LEC}$ mice as compared with controls (body weight in female $Cd36^{\Delta LEC}$ mice was $29.1 \pm 0.4$ g vs $27 \pm 0.4$ g in controls, $P = 0.016$, and in male $Cd36^{\Delta LEC}$ mice

$33.6 \pm 0.5$ g vs $26.5 \pm 0.2$ g in controls, $P < 0.01$) (Fig. 3a). At 20 weeks, body composition assessed by DEXA showed an increase in fat mass in both female and male $Cd36^{\Delta LEC}$ mice as compared with their respective controls (female: $8.7 \pm 0.2$ g vs $6.5 \pm 0.3$ g; male: $12.3 \pm 0.5$ g vs $6.5 \pm 0.1$ g; all $P < 0.001$) with no difference in lean mass between groups (female $P = 0.327$; male $P = 0.154$) (Fig. 3b). The increase in weight involved more accumulation of epididymal fat ($P = 0.0033$) and also of subcutaneous fat ($P < 0.001$) (Fig. 3c, d).

Oral glucose tolerance (OGTT) test showed no difference in glucose disposal rate in 11-week-old $Cd36^{\Delta LEC}$ and control mice (Fig. 3e), whereas glucose disposal was impaired in $Cd36^{\Delta LEC}$ mice at 20 weeks as compared with the matched controls ($P < 0.01$) (Fig. 3f). The epididymal adipose tissue was examined for expression of key adipogenic and inflammatory markers in 11- and 20-week-old $Cd36^{\Delta LEC}$ and the respective controls (Fig. 3g). At 11 weeks, tissues of $Cd36^{\Delta LEC}$ and control mice showed no differences in gene expression of the lipid storage genes peroxisome proliferator-activated receptor gamma ($Pparg$) and lipoprotein lipase ($LpL$). Expression of pro-inflammatory interleukin-6 ($Il6$), and of pro-fibrotic tumor growth factor-beta

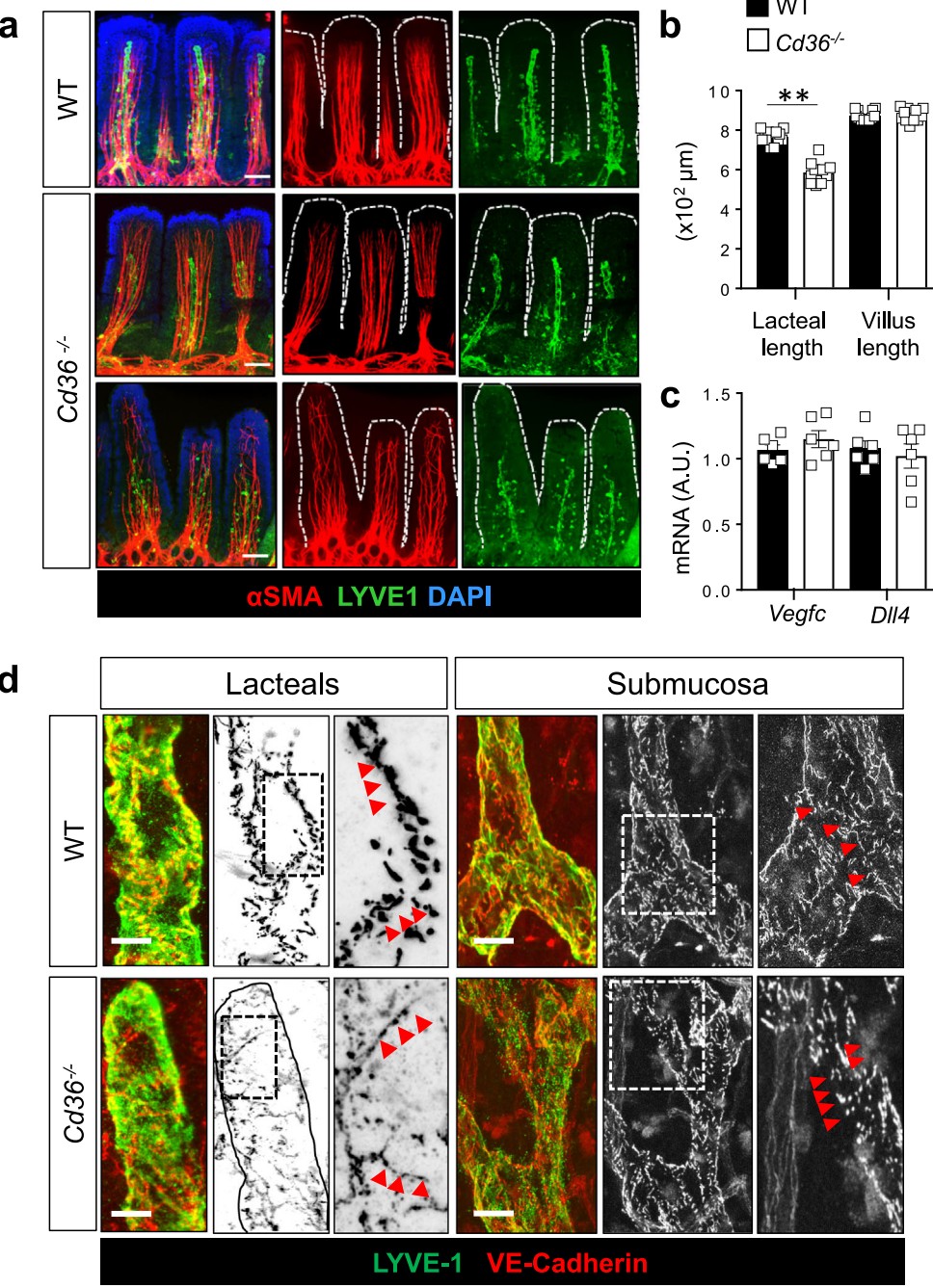

**Fig. 2 *Cd36*$^{-/-}$ mice have shorter lacteals and disorganized VE-cadherin junctions. a** Representative whole-mount staining for lymphatic vessel endothelial hyaluronan receptor 1, LYVE-1, and smooth muscle actin-alpha, αSMA, showing lacteals in jejunum of wild-type (WT) and *Cd36*$^{-/-}$ mice. Images were acquired using a Zeiss LSM 880 Airyscan Confocal Microscope, ×20 objective. Scale bar: 50 μm. **b** Lacteal and villus length quantified using ImageJ v1.53i software. *Cd36*$^{-/-}$ mice lacteal length is reduced as compared to WT ($P < 0.01$) **c** mRNA expression of vascular endothelial growth factor C, *Vegfc*, and of delta-like ligand 4, *Dll4*, in jejunum of WT and *Cd36*$^{-/-}$ mice. **d** Immunohistochemical analysis performed in whole-mount jejunum stained for vascular endothelial (VE)-cadherin (red) and LYVE-1 (green) showing disorganized VE-cadherin junctions in *Cd36*$^{-/-}$ lacteal and in collecting lymphatic vessels as compared with WT mice. Images were acquired using a Zeiss LSM 880 Airyscan Confocal microscope, ×63 objective. Scale bar is 5 μm. Mice ($n = 6$, representative of two independent experiments) used in (**a–b**) are 12-week old. All data are means ± SE with *n* representing the number of mice per group. A.U. arbitrary units. Statistical significance is determined by two-tailed Student *t* test. **$P < 0.01$.

(*Tgfb1*) was also similar. However, the expression of genes encoding for pro-inflammatory cytokine *Tnf*, tumor necrosis factor alpha, and the macrophage marker EGF module-containing mucin-like receptor 1 (*Emr1*) were increased ($P < 0.05$ and $P < 0.05$, respectively) as compared with age-matched control mice (Fig. 3g). At 20 weeks, *Cd36*$^{ΔLEC}$ mice showed ~3-fold increases in *LpL*, *Pparg*, *Tnf*, and *Il6* gene expression and

~7-fold increase in *Tgfb1* and *Emr1* gene expression (all $P < 0.001$) (Fig. 3g). These data suggested that inflammation of visceral fat precedes the increase in lipid storage genes, the expansion of the tissue and the onset of glucose intolerance. We evaluated energy expenditure in 11- and 20-week-old *Cd36*$^{ΔLEC}$ mice and their respective controls. The 11-week-old *Cd36*$^{ΔLEC}$ mice showed decreased VCO$_2$ (dark and light periods) (Fig. 4a),

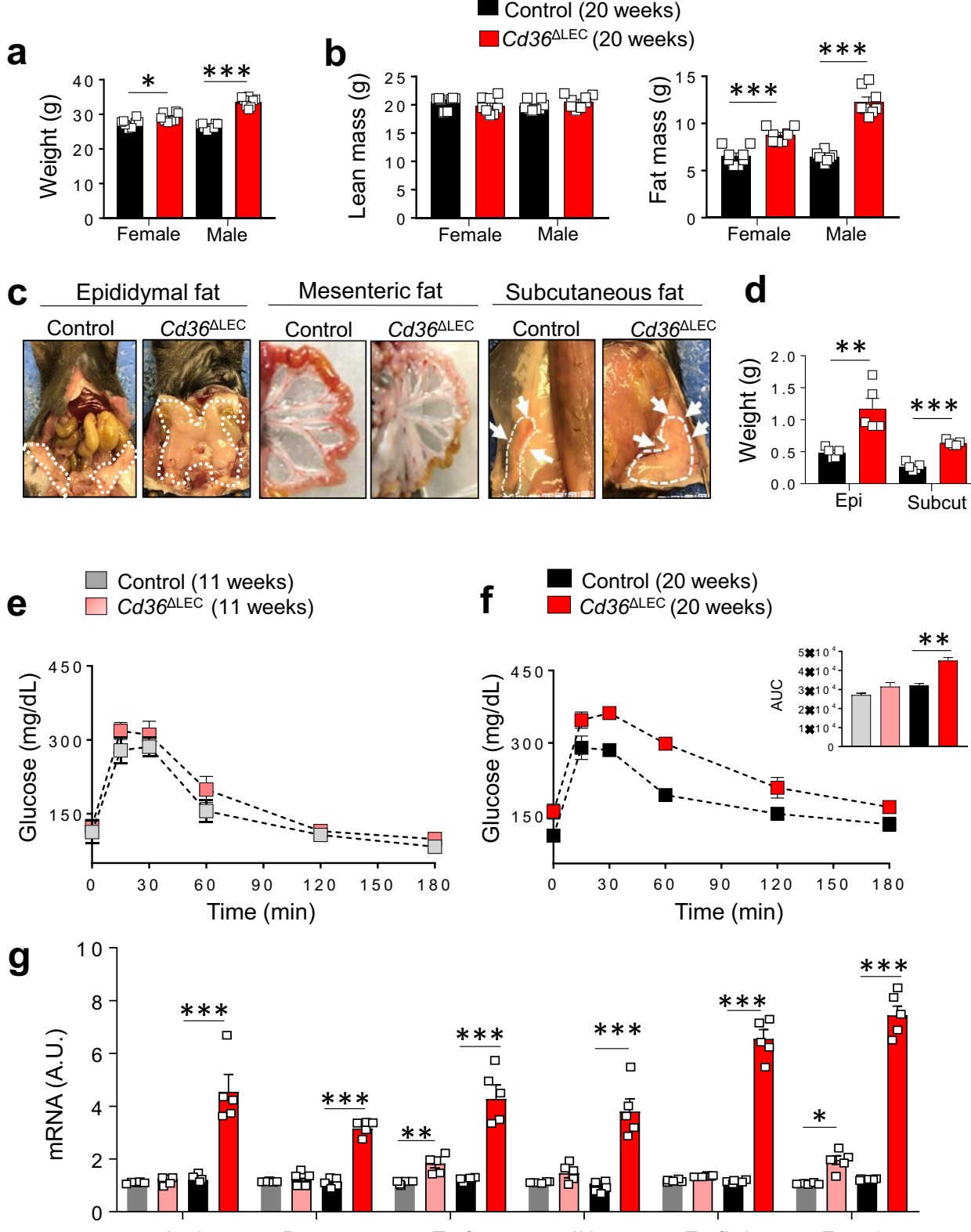

VO$_2$, and energy expenditure (light period) (Fig. 4b, c) (all $P < 0.05$) and these differences became even more significant at 20 week of age (Fig. 4d–f) (all $P < 0.01$).

The increase in visceral adiposity in $Cd36^{\Delta LEC}$ mice suggested the possibility that lymphatic integrity might be compromised. Previously, mice lacking one allele of the homeobox gene Prox1[7], the transcription factor important for LEC identity[6] were shown

to accumulate more visceral fat. We investigated the organization of LEC VE-cadherin junctions in whole-mount intestines by immunohistochemistry and found more fragmented VE-cadherin staining in lacteals (Fig. 5a) and submucosa of $Cd36^{\Delta LEC}$ mice as compared with those in control mice (Fig. 5b). This indicated that deletion of CD36 in LECs is sufficient to disrupt the morphological status of VE-cadherin at cellular junctions. As

**Fig. 3 $Cd36^{\Delta LEC}$ mice develop spontaneous obesity and adipose tissue inflammation. a** Body weight (g) of female and male 20-week LEC $Cd36$ deficient ($Cd36^{\Delta LEC}$) mice is increased as compared with sex-matched control mice ($n = 8$); female, $P = 0.016$; male, $P < 0.01$. **b** Body composition analysis shows similar lean mass between age- and sex-matched controls and $Cd36^{\Delta LEC}$ mice (female, $P = 0.327$; male, $P = 0.154$), whereas fat mass is higher in both female and male $Cd36^{\Delta LEC}$ mice (both $P < 0.01$) as compared with sex-matched controls ($n = 8$). **c** Representative images of visceral (epididymal and mesenteric) and subcutaneous fat pads from male mice. **d** Weight (g) of epididymal (Epi) and subcutaneous (Subcut) fat pads is increased (all $P < 0.01$) in $Cd36^{\Delta LEC}$ mice as compared with controls ($n = 5$). **e, f** Levels of plasma glucose and area under the curve (AUC) during an oral glucose tolerance test in 11- and 20-week-old mice ($n = 5$). Glucose disposal is impaired in $Cd36^{\Delta LEC}$ mice at 20 weeks as compared with age-matched controls ($P < 0.01$). **g** Gene expression analysis in epididymal adipose tissue in 11- and 20-week-old $Cd36^{\Delta LEC}$ and age-matched control mice ($n = 5$). As compared with age-matched controls, 11-week-old $Cd36^{\Delta LEC}$ mice show increased gene expression of tumor necrosis factor alpha ($Tnf$, $P < 0.01$) and of EGF module-containing mucin-like receptor 1 ($Emr1$, $P < 0.05$). The 20-week-old $Cd36^{\Delta LEC}$ mice show increased gene expression of lipoprotein lipase ($LpL$, $P < 0.001$), peroxisome proliferator-activated receptor gamma ($Pparg$, $P < 0.001$), $Tnf$ ($P < 0.001$) interleukin-6 ($Il6$, $P < 0.001$), tumor growth factor-beta ($Tgfb1$, $P < 0.001$) and $Emr1$ ($P < 0.001$). A.U. arbitrary units. All data are means ± SE with $n$ representing the number of mice per group. Statistical significance is determined by two-tailed Student $t$ test. *$P < 0.05$; **$P < 0.01$, ***$P < 0.001$.

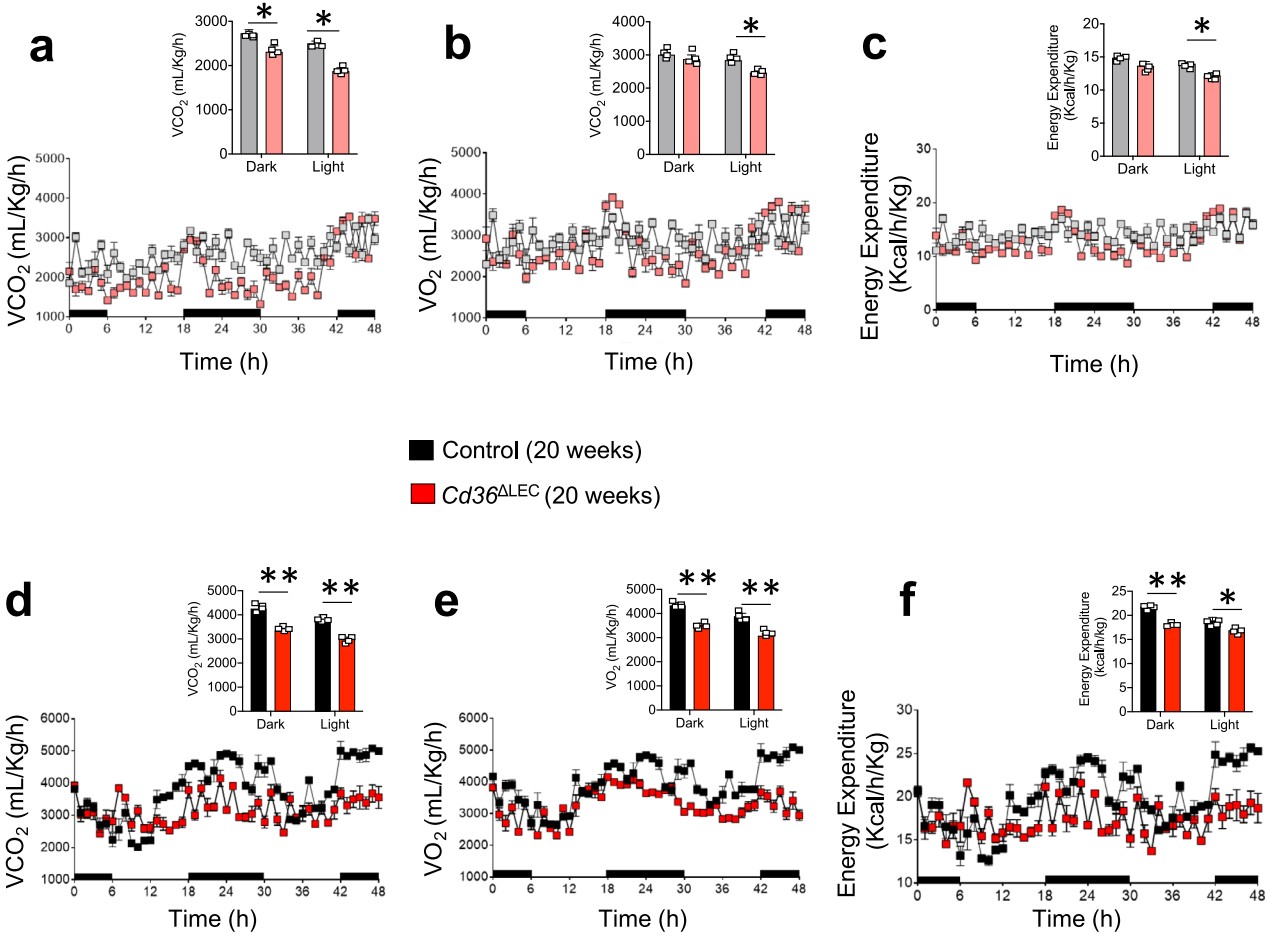

**Fig. 4 $Cd36^{\Delta LEC}$ mice have reduced energy balance.** Eleven- and 20-week-old $Cd36^{\Delta LEC}$ and control mice ($n = 4$) were housed in metabolic cages for 48 h to monitor **a**, **d** carbon dioxide release rate (VCO$_2$), **b**, **e** O$_2$ consumption (VO$_2$), and **c**, **f** energy expenditure. Data are means ± SE with $n$ representing the number of mice per group. Light grey: 11-week-old control mice; light red: 11-week old $Cd36\Delta LEC$ mice. Statistical significance is determined by two-tailed Student $t$ test. *$P < 0.05$; **$P < 0.01$.

expected, no differences were observed in VE-cadherin junction morphology for blood vessels surrounding the lacteals between the two mouse groups (Supplementary Fig. 4a).

Collecting lymphatic vessels have continuous inter-endothelial VE-cadherin junctions optimal for lymph transport[34] and disruption of vessel integrity can result in lymph leakage, as has been reported in mice lacking one allele of Prox1[7]. We examined lymph transport by lymphatic vessels in the mesentery collecting vessels of $Cd36^{\Delta LEC}$ mice by fluorescence microscopy after an intragastric load of the green fluorescent long-chain fatty acid tracer BODIPY C16[7]. While BODIPY highlighted lymph flow and integrity of lymphatic transport in control mice (Fig. 5c), BODIPY leakage from lymphatic vessels in the mesentery was evident in $Cd36^{\Delta LEC}$ mice (Fig. 5c). We next examined how lymph leakage in $Cd36^{\Delta LEC}$ mice impacts lymph transport of absorbed lipid to the circulation. The $Cd36^{\Delta LEC}$ and control mice received an intragastric bolus of olive oil after injection of Triton WR 1339 to inhibit lipoprotein lipase and rapid clearance of plasma triglycerides (TGs)[31]. In control mice, plasma TG levels increased at 1 h ($600 \pm 90$ mg/dl) following the oral fat bolus, peaked at 3 h ($846 \pm 151$ mg/dl), decreased at 5 h ($452 \pm 139$ mg/dl) and returned back to pre-meal levels after 7 h ($166 \pm 28$ mg/dl).

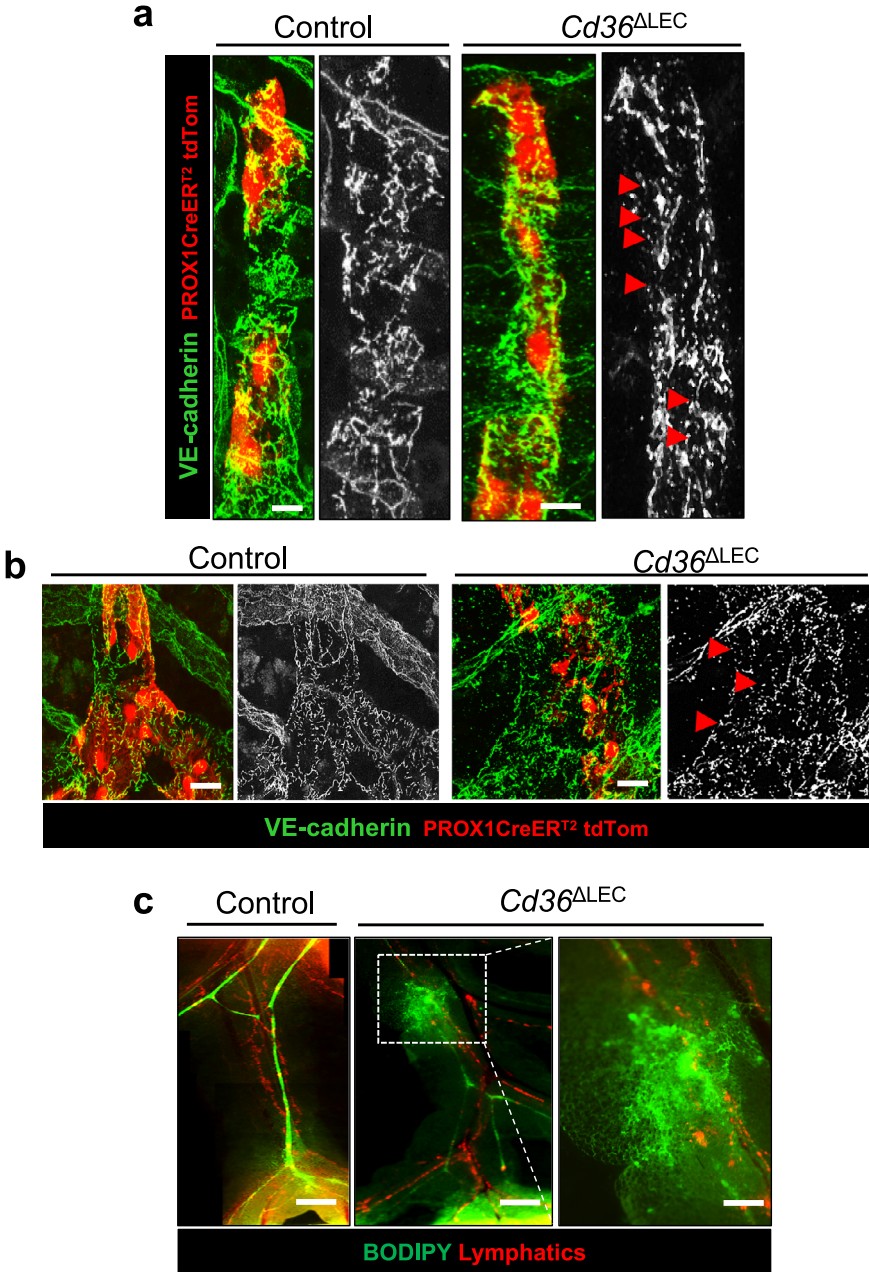

**Fig. 5 Cd36<sup>ΔLEC</sup> mice have more discontinuity of LEC VE-cadherin junctions and leaky mesenteric lymphatic vessels.** Representative images of whole-mount VE-cadherin (green) immunohistochemistry in (**a**) lacteals and (**b**) submucosa collecting vessels in 20-week-old Cd36<sup>ΔLEC</sup> and control mice (n = 5). TdTomato (TdTom, red) identifies PROX1 positive cells. Images were acquired in tiled scans using a Zeiss LSM 880 Airyscan Confocal Microscope, ×40 objective. Scale bar is 5 μm. **c** Fluorescent long-chain fatty acid tracer BODIPY C16 was administered intragastrically and integrity and transport function of lymphatic vessels recorded using a stereomicroscope. Scale bars: 500 μm. Right panel shows high-magnification image of the area outlined by the white dashed box (Scale bar: 200 μm). Images are representative of two independent experiments with n representing the number of mice per group.

In contrast, as compared with controls, plasma TG levels in Cd36<sup>ΔLEC</sup> mice were ~50% lower at 1 h (P < 0.01), maintained a plateau between 3 h, and remained higher at 5 and 7 h (P < 0.05 and P < 0.001 respectively). (Supplementary Fig. 4b). Thus, lipid absorption was delayed and prolonged in Cd36<sup>ΔLEC</sup> mice in line with the disruption of lymphatic transport. In summary, the data show that CD36 deletion in LECs of adult mice disrupts lymphatic VE-cadherin junctions and increases lipid permeability of mesenteric lymphatic vessels. Lipid transport to the circulation is delayed which prolongs the postprandial TG phase. The

Cd36<sup>ΔLEC</sup> mice accumulate visceral adipose tissue that is inflamed and display glucose intolerance.

Disruption of lymphatic vessel integrity[8,9] has been reported in mice models of obesity and T2D. We examined if diet-induced obesity affects LEC CD36 expression in WT mice fed a high-fat diet (HFD) for 12 weeks. The HFD did not affect CD36 protein expression in mucosa LECs (lacteals) as compared to chow diet condition (P = 0.36), but it increased CD36 expression in LECs from submucosa and mesenteric lymphatic vessels as compared to gut LECs from mice fed chow diet (P < 0.01 and P < 0.001

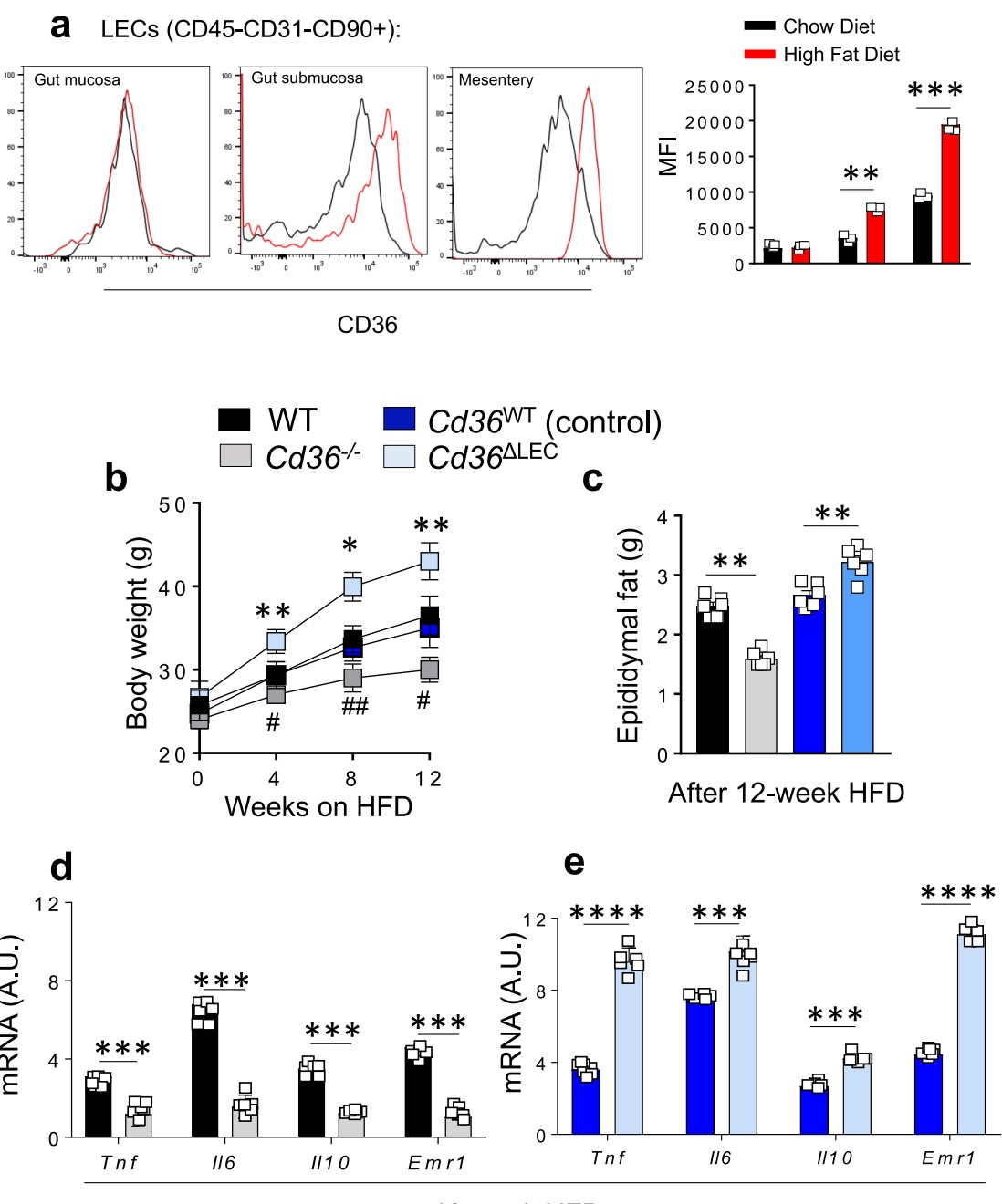

**Fig. 6 High-fat diet feeding induces obesity and adipose tissue inflammation in $Cd36^{\Delta LEC}$ mice but not in $Cd36^{-/-}$ mice. a** Eight-week-old male and female wild-type (WT) mice ($n = 3$) were fed either chow diet (CD) or high-fat diet (HFD) for 12 weeks. CD36 expression was measured by flow cytometric analysis in lymphatic endothelial cells (LECs) (CD45−CD90.2+CD31+) isolated from gut mucosa, submucosa, and mesentery lymphatic vessels. CD36 expression is increased in LEC isolated from gut submucosa and mesenteryic vessels ($P < 0.01$ and $P < 0.001$, respectively). MFI: mean fluorescence intensity. **b** Body weight of $Cd36^{-/-}$ and $Cd36^{\Delta LEC}$ mice, and respective controls ($n = 6$), fed HFD for 12 weeks. *Depicts significance between $Cd36^{\Delta LEC}$ and $Cd36^{WT}$ mice (4 weeks, $P < 0.001$; 8 weeks, $P < 0.01$; 12 weeks, $P < 0.001$)#, depicts significance between WT vs $Cd36^{-/-}$ mice (4 weeks, $P < 0.05$; 8 weeks, $P < 0.01$; 12 weeks, $P < 0.05$). **c** Epidydimal fat pad weight (g) is decreased in $Cd36^{-/-}$ mice as compared to WT mice ($P < 0.001$), whereas epidydimal fat pad weight is increased in $Cd36^{\Delta LEC}$ mice as compared to controls ($P < 0.01$) fed HFD for 12 weeks. **d** Cytokine gene expression is decreased in $Cd36^{-/-}$ epidydimal adipose tissue as compared to WT mice ($n = 6$) fed HFD for 12 weeks. **e** Cytokine gene expression is increased in $Cd36^{\Delta LEC}$ epidydimal adipose tissue as compared to controls ($n = 6$) fed HFD for 12 weeks. Tumor necrosis factor alpha, *Tnf*; Interleukin-6, *Il6*; Interleukin-10, *Il10*; EGF module-containing mucin-like receptor 1, *Emr1*. A.U. arbitrary units. All data are from two independent experiments and are means ± SE with *n* representing the number of mice per group. Statistical significance is determined by two-tailed Student *t* test. * and #$P < 0.05$; **$P < 0.01$; ***, $P < 0.001$; ****$P < 0.0001$.

respectively) (Fig. 6a). In addition, $Cd36^{-/-}$, $Cd36^{\Delta LEC}$ mice and their controls were included in the HFD regimen to compare effects on body weight, epididymal fat mass, and markers of inflammation. The HFD resulted increases body weight and epididymal fat mass in $Cd36^{\Delta LEC}$ mice as compared to controls (4 weeks, $P < 0.001$; 8 weeks, $P < 0.01$; 12 weeks, $P < 0.001$), while germline $Cd36^{-/-}$ mice, did not gain as much body weight, as compared to WT mice (4 weeks, $P < 0.05$; 8 weeks, $P < 0.01$; 12 weeks, $P < 0.05$) (Fig. 6b, c), as previously reported[37]. HFD associated with increased expression of genes encoding for key proinflammatory cytokines in epididymal fat of WT and control mice and these genes showed further upregulation in epididymal fat of $Cd36^{\Delta LEC}$ mice ($P < 0.001$ and $P < 0.0001$). The $Cd36^{-/-}$ mice accumulated less epididymal fat as compared to WT mice ($P < 0.001$) and displayed low adipose tissue inflammation documented by decreased expression of key pro-inflammatory cytokines (all $P < 0.001$) (Fig. 6d, e). These data show that HFD increases CD36 expression in collecting lymphatics, which would optimize vessel transport of the absorbed fat and chronically could contribute to weight gain.

**CD36 regulates LEC oxidative metabolism and function in vitro**. To understand if the compromised lymphatic integrity observed in $Cd36^{\Delta LEC}$ mice can be related to autonomous CD36 regulation of LECs, we examined the impact of CD36 silencing in human dermal LEC function in vitro. LECs have robust CD36 expression (Fig. 7a) and a 60–80% reduction of CD36 protein content was achieved using two different CD36 siRNAs (Fig. 7a, b) ($P < 0.0001$). CD36 silencing in LECs did not affect expression of genes encoding the fatty acid transporters FATPs, namely $SLC27A1$, $SLC27A3$, and $SLC27A4$, as compared with control (Ctrl) siRNA LECs (Fig. 7c). HDLECs have negligible expression of $SLC27A5$ and $SLC27A6$ as previously reported[38]. LEC function relies on fatty acid β-oxidation (FAO)[29], a pathway shown to be regulated by CD36 in several cell types[39–43]. CD36 silencing lowered by ~50% basal respiration measured by oxygen consumption rate (OCR) as compared with Ctrl siRNA LECs ($P < 0.05$) whereas ATP-linked and maximal OCR did not differ. VEGF-C treatment (16 h) increased basal, ATP-linked OCR, and maximal OCR ($P < 0.01$ and $P < 0.001$) in Ctrl siRNA LECs. However, the effect of VEGF-C on cell respiration was largely prevented by CD36 silencing for all three OCR measurements (all $P < 0.01$) (Fig. 7d, e). We examined the effect of CD36 silencing on glucose utilization by assessing extracellular acidification rate (ECAR), which measures lactate production. In control siRNA LECs, VEGF-C significantly increased glycolysis and glycolytic rates as compared to untreated cells (both $P < 0.001$). In contrast, CD36 siRNA LECs had substantially increased glycolysis under basal condition (all $P < 0.01$) that did not respond to VEGF-C stimulation (Fig. 7f, g). In line with the above data, CD36 silencing in VEGF-C treated LECs significantly reduced gene expression of key enzymes of FAO, namely carnitine palmitoyl-transferase 1, $Cpt1a$ ($P < 0.001$), long-chain fatty acid-CoA ligase 1, $Acsl1$ ($P < 0.01$) as previously reported[44], and very long-chain acyl-CoA dehydrogenase, $Vlcad$ ($P < 0.001$), as previously reported[45] (Fig. 7h) while it increased gene expression of glycolysis related genes, glucose transporter 1 ($GLUT1$) ($P < 0.0001$), hexokinase 2 ($HK2$) ($P < 0.05$) and aldolase A ($ALDOA$) ($P < 0.001$) (Fig. 7i). The same glucose utilization genes were increased by VEGF-C treatment in Ctrl siRNA LECs ($P < 0.01$), as previously reported[46], but did not respond to VEGF-C in CD36 siRNA LECs. Expression of 6-phosphofructo-2-kinase/fructose-2,6-biphosphatase 3 ($PFKFB$) was unaltered by CD36 silencing or VEGF-C treatment (Fig. 7i). In vivo relevance of these findings was shown by measuring decreased mRNA of $Cpt1a$ ($P < 0.01$), $Acsl1$ ($P < 0.05$) and $Vlcad$ ($P < 0.01$) and increased mRNA of $Glut1$ ($P < 0.001$) in LECs sorted from 20-week-old $Cd36^{\Delta LEC}$ mice as compared to those from controls (Fig. 7j).

CD36 siRNAs reduced cell migration (both $P < 0.001$) as compared with Ctrl siRNA LECs (Fig. 8a, b). VEGF-C induced tube formation was decreased only by CD36 siRNA1 (branch points $P < 0.001$; tube length $P < 0.001$) (Fig. 8c). We next examined whether CD36 silencing impacts the morphology of VE-cadherin junctions in LECs. VEGF-C treatment of 6-h serum-starved LECs induced visible smoothening of VE-cadherin into continuous, linear junctions but this effect was diminished in CD36 siRNA LECs (Fig. 8d). A similar pattern was observed with VEGF-A treatment (Fig. 8d). VE-cadherin protein levels were not changed by CD36 silencing (Fig. 8e). In addition to being regulated by VEGF-C, the status of VE-cadherin itself influences VEGFR2 and AKT signaling[47]. We examined if the attenuated actions of VEGF-C in siRNA CD36 LECs reflect impaired VEGF-C signaling. As compared with Ctrl siRNA LECs, VEGF-C signaling to VEGFR2 and AKT was reduced in CD36 siRNA LECs as phosphorylation of VEGFR2 at $Y^{1175}$ and of AKT at $S^{473}$ ($P < 0.05$) was suppressed (Fig. 8f, g). Plasma membrane CD36 level was not changed by VEGF-C treatment or serum omission suggesting that CD36 cycling[48–50] is unaltered (Supplementary Fig. 5).

## Discussion

The current study increases our understanding of the regulation of intestinal lymphatics by documenting the role of the fatty acid transporter CD36 in the maintenance of lymphatic vessel integrity. The findings mechanistically explain our previous observations that $Cd36^{-/-}$ mice have reduced lipid secretion into the cannulated mesenteric lymph duct[31]. We show that CD36 is abundantly expressed in intestinal LECs, although not all Lyve-1 expressing cells have CD36, which is in line with the reported heterogeneity of gene expression and function of LECs[33]. We describe an increasing CD36 expression gradient from lacteals to lymphatic vessels in submucosa and mesentery and an effect of high-fat diet to upregulate CD36 level in LECs isolated from collecting lymphatic vessels. Germline deletion of CD36 ($Cd36^{-/-}$) in mice reduced length of the lacteals and impaired organization of VE-cadherin junctions, which appeared fragmented and reduced in density. The shorter lacteals in $Cd36^{-/-}$ mice do not reflect reduced proliferation as this was likely sustained by the increase in basal glycolysis observed in CD36 deficient LECs. Glycolytic rates in LECs are much higher than FAO rates[51] and fuel cellular proliferation and migration, while FAO plays a role in promoting transcription of key lymphangiogenic genes[29]. Shorter lacteals were not observed in $Cd36^{\Delta LEC}$ mice suggesting that potential differences in gradient of angio- and lymphangiogenic growth factors, important for LEC migration and tube formation, in addition to the reduced VEGF-C mediated signaling, might have caused the shorter lacteals in $Cd36^{-/-}$ mice.

The $Cd36^{\Delta LEC}$ displayed more discontinuity of VE-cadherin junctions and lymph leakage from collecting lymphatics following intragastric administration of a fluorescent lipid tracer. The mice developed spontaneous visceral and also subcutaneous adiposity, inflammation of the visceral fat and systemic glucose intolerance (Fig. 9). Together these data suggest that the primary function of LEC CD36 is to optimize lymphatic vessel integrity and lymph transport of dietary fat.

The lymphatic dysfunction in $Cd36^{\Delta LEC}$ mice reflected autonomous regulation by CD36 in LECs. Depletion of CD36 in LECs from $Cd36^{\Delta LEC}$ reduced the expression of genes encoding key enzymes of FAO such as $CPT1A$ and $ACSL1$ while increasing expression of those related to glucose utilization such as $Glut1$. CD36 silencing (CD36siRNA LECs) reduced cell respiration but significantly increased basal glycolytic rate and expression of the key glucose utilization genes $GLUT1$ and $HEK2$. Both FAO and glycolysis are important for lymphatic maintenance and are enhanced by VEGF-C to fuel cell migration, and tube sprouting

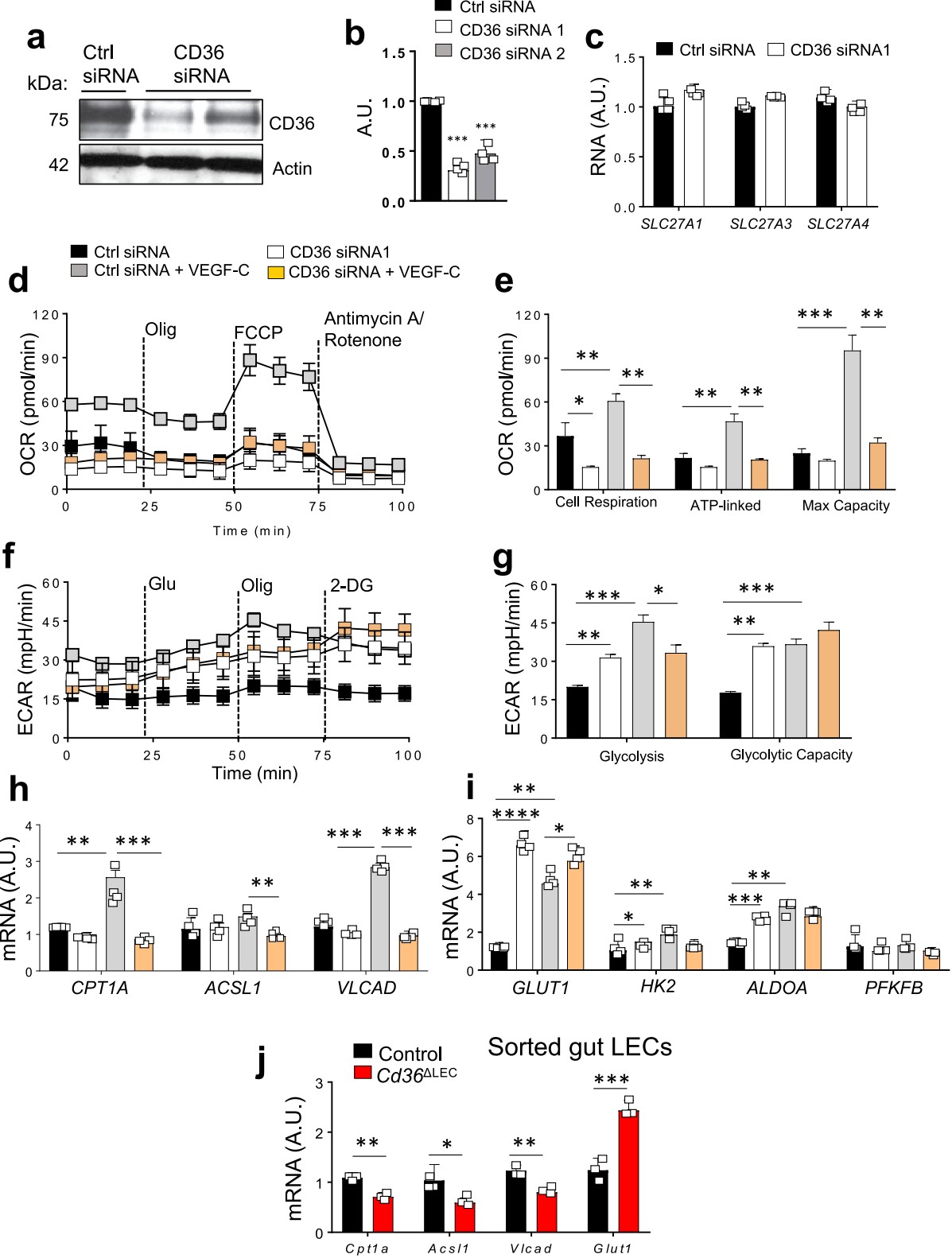

during lymphangiogenesis[29,52]. CD36 silencing reduced VEGF-C action to upregulate gene expression of *GLUT1*, *HEK2,* and *CPT1A* and its ability to stimulate cell respiration, glycolysis, cell migration, and tube formation. The defects in tube formation and migration appear mediated by VEGF-C signaling through VEGFR2 and AKT; VEGF-C activation of VEGFR2$^{Y1175}$ and AKT$^{S473}$ was inhibited by CD36 silencing. The VEGFR2 pathway regulates the morphology of VE-cadherin junctions in LECs[2], and AKT activity independently contributes to junction stabilization[53]. VEGF-C treatment did not optimize continuity of VE-cadherin junctions in CD36 depleted LECs as it did in control LECs.

Consistent with the LEC autonomous effect of CD36 deficiency to increase discontinuity of VE-cadherin junctions shown by the

**Fig. 7 CD36 regulates lymphatic endothelial cell oxidative metabolism.** Human dermal lymphatic endothelial cells (LECs) transfected at 40–60% confluence with either control (Ctrl siRNA) or CD36 Silencer® Pre-designed siRNA (CD36 siRNA1 and 2) are assayed at 95–100% confluence. **a, b** CD36 knockdown evaluated by western blotting and quantified ($n = 4$ per condition) ($P < 0.001$). A.U. arbitrary units, kDa kilodalton. **c** Expression of genes encoding for fatty acid transporters (*Slc27a1, Slc27a3,* and *Slc27a4*) in Ctrl and CD36 siRNA LECs ($n = 4$ per condition). **d, e** LEC oxygen consumption rate (OCR) profile and its quantification by Seahorse (*$P < 0.05$; **$P < 0.01$; ***$P < 0.001$). **f, g** Extracellular acidification (ECAR) rate and its quantification by Seahorse (**$P < 0.01$; ***$P < 0.001$). **e, g** (quantification: each bar represents 6 wells/condition averaged for three time points, $n = 18$) are representative of two independent experiments. **h–i** Expression of genes encoding for key enzymes of fatty acid β-oxidation (FAO) and glycolysis in Ctrl and CD36 siRNA LECs with/without VEGF-C treatment (16 h). Carnitine palmitoyltransferase 1a, *CPT1a;* Long-chain fatty acid CoA ligase 1, *ACSL1;* Very long-chain acyl-CoA dehydrogenase, *VLCAD;* Glucose transporter 1, *GLUT1;* Hexokinase 2, *HK2;* Aldolase A, *ALDOA;* 6-phosphofructo-2-kinase/fructose-2,6-biphosphatase 3, PFKFB. (j) Expression of key FAO genes *Cpt1a* ($P < 0.01$), *Acsl1* ($P < 0.05$), *Vlcad* ($P < 0.01$), and of Glucose transporter 1, *Glut1* ($P < 0.001$) in LECs sorted from 20-week-old $Cd36^{\Delta LEC}$ and control mice ($n = 3$). A.U. arbitrary units. Data are means ± SE with $n$ representing the number of mice per group. Statistical significance is determined by two-tailed Student $t$ test. *$P < 0.05$; **$P < 0.01$; ***$P < 0.001$; ****$P < 0.0001$.

---

in vitro data, both $Cd36^{-/-}$ and $Cd36^{\Delta LEC}$ mice displayed more discontinuous VE-cadherin junctions in intestinal whole mounts as compared with respective controls. $Cd36^{\Delta LEC}$ mice maintained on a HFD accumulated more fat mass and had exacerbated inflammation of epididymal fat while the $Cd36^{-/-}$ mice were protected. The differences in adiposity between these mice models likely reflect several factors. $Cd36^{\Delta LEC}$ mice express CD36 on blood endothelial cells as well as on adipocytes, while global CD36 loss in $Cd36^{-/-}$ mice suppresses blood endothelial FA delivery to tissues and mitigates diet-induced obesity[18,37]. Lacteal LEC junctions were highly fragmented in $Cd36^{-/-}$ mice and when intragastric BODIPY C16 was administered we observed a rapid increase of fluorescence levels in the circulation. This is in line with previous findings in $Cd36^{-/-}$ mice where impaired lipid secretion into the cannulated mesenteric lymph duct, abnormally rapid blood appearance of absorbed TG[31], and reduced integrity of blood endothelial vessels[17] were observed. Thus, in $Cd36^{-/-}$ mice chylomicrons likely reach the circulation directly and independently of lymphatic transport. By comparison, lacteal junctions in $Cd36^{\Delta LEC}$ mice although more discontinuous were not as fragmented as in $Cd36^{-/-}$ mice, so lymphatic lipid transport occurred but was defective resulting in lymph leakage with onset of inflammation and obesity.

Our findings in mice and cultured LECs suggest that a major functional consequence of LEC CD36 deficiency is impairment of lymphatic vessel barrier integrity. The mesenteric lymph vessels of $Cd36^{\Delta LEC}$ mice leaked the fluorescent long-chain fatty acid tracer BODIPY C16, and these vessels are closely associated with abdominal fat depots, which contributes to the expansion of mesenteric and visceral adipose depots, as previously observed in mice with Prox1 haploinsufficiency[7]. Fatty acids from the lymph were shown to promote proliferation and differentiation of pre-adipocytes and the hypertrophy of adipocytes[54]. We also measured enhanced gene expression of *LpL* in visceral adipose tissue of $Cd36^{\Delta LEC}$ mice, which could augment fatty acid availability from chylomicrons present in the leaked lymph. The accumulated visceral fat in $Cd36^{\Delta LEC}$ mice had several fold increases in gene expression of proinflammatory markers *Tnf* and *Il6*, of the pro-fibrotic *Tgfb1* and of the macrophage marker *Emr1* that associates with inflamed adipose tissue[55,56]. These proinflammatory changes are likely consequent to the leaked lymph diverting to adipose tissue immune cells that would have normally trafficked to lymph nodes[57]. In support of this, we found significant upregulation of *Tnf* and *Emr1* gene expression in the adipose tissue of 11-week-old $Cd36^{\Delta LEC}$ mice before upregulation of the lipid storage genes *LpL* and *Pparg*, observed later in 20-week-old $Cd36^{\Delta LEC}$ mice. Thus, inflammation in adipose tissue precedes the onset of obesity and glucose intolerance in $Cd36^{\Delta LEC}$ mice. Chronic inflammation of visceral adipose tissue is linked to the etiology of systemic insulin resistance[58–60], as pro-inflammatory mediators, particularly IL-6, reach the liver through the portal system which

drains blood from the visceral fat depot[61]. In addition, we observed a small reduction in energy expenditure in 11-week-old $Cd36^{\Delta LEC}$ mice, as compared with controls, which might have contributed to the weight increase. We cannot exclude contribution of other organs[62–64] to some of the phenotypes observed in $Cd36^{\Delta LEC}$ mice. The decrease in energy expenditure might be driven by alterations in intestinal homeostasis, namely inflammation and/or changes in the microbiota[65].

In summary, our findings connect integrity and function of gut/mesenteric lymphatic to expression of CD36, a protein with regulatory influence in energy metabolism[13]. It is interesting to note the parallel between the function of LEC CD36 which optimizes transport of absorbed lipid from the intestine to the circulation for distribution to tissues, and that of blood endothelial cell CD36 which optimizes cellular uptake of FAs from the circulation to tissue parenchymal cells[18]. This suggests that CD36 regulation of nutrient bioavailability is not solely defined by its function in cellular FA uptake.

Deficiency of CD36 in LECs highlights a new mechanism for the etiology of visceral obesity and insulin resistance, two phenotypes linked to the metabolic syndrome, which increases risk of cardiovascular disease and T2D, pathologies that associate with *CD36* genetic variants[66–68]. Genetically determined low *CD36* mRNA associates with incidence of T2D and its metabolic complications in genome-wide/RNA sequencing data[68] and vascular endothelial dysfunction is observed in individuals carrying the minor allele of a coding *CD36* single nucleotide polymorphism[69]. Considering the important function of the lymphatic system in tissue homeostasis, LEC CD36 might also contribute to disease association.

## Methods

**Mice and metabolic studies.** Female and male C57BL/6 wild-type (WT) and *Cd36*-null ($Cd36^{-/-}$) mice[24] were used at 12–14 weeks. Sex- and age-matched WT mice were used as controls of $Cd36^{-/-}$ mice. Male and female mice with *Cd36* deletion in LECs were obtained by crossing Prox1-CreER$^{T2}$-tdTomato reporter mice[36] with $Cd36^{fl/fl}$ mice[17,18] to generate Prox1-CreER$^{T2}$-tdTomato$Cd36^{fl/fl}$ (referred to as $Cd36^{\Delta LEC}$). Flox negative, Cre positive littermate mice were used as controls (referred to as controls). All mice were on a C57BL/6 background and were bred in the same facility. Cre-mediated deletion was induced in 8-week-old mice by intragastric administration of tamoxifen (20 mg/ml) in corn oil (Sigma Aldrich) three times/week for two weeks. Littermate controls received the same tamoxifen protocol. High-fat diet studies were conducted in 8-week-old male and female WT, $Cd36^{-/-}$, $Cd36^{\Delta LEC}$ and control mice fed 60% kcal Fat (Research Diet Inc. Cat# D12492) for 12 weeks. All mice were housed in a 12 h light-dark cycle, and at a temperature of 65–75 °F (18–23 °C) with 40–60% humidity, in full-barrier facilities.

Whole-body fat and lean mass were assessed with an EchoMRI Whole Body Composition AnalyzerBody (Echo Medical Systems). For oral glucose tolerance test (OGTT), intragastric glucose (2 g/kg, Sigma-Aldrich) was given to mice fasted for 6 h and tail blood collected at 0, 15, 30, 60, 120, and 180 min for glucose measurements. Single-chamber indirect calorimetry system (Columbus Instruments) was used for dark and light periods for 48 h in the fed state using chow diet. $O_2$ consumption ($VO_2$), $CO_2$ production ($VCO_2$), and energy expenditure were measured for 48 h. All animal studies were performed in accordance with the guidelines provided by the National Institute of Health. All

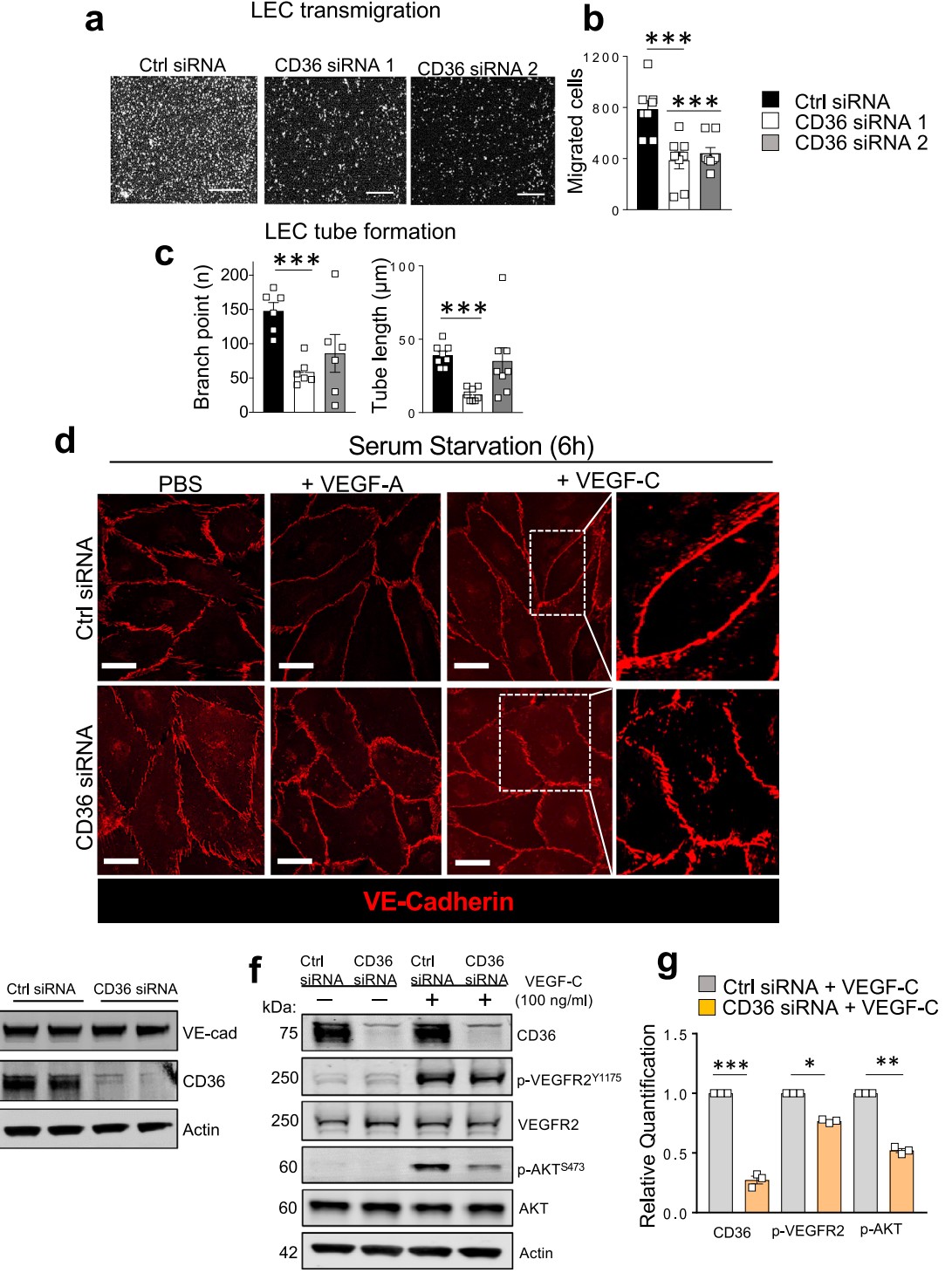

**Fig. 8 CD36 deletion impairs lymphatic endothelial cell function, VE-cadherin morphology, and VEGFR2/AKT activation.** Human dermal lymphatic endothelial cells (LECs) are transfected with either control (Ctrl siRNA) or CD36 Silencer® Pre-designed siRNA (CD36 siRNA1 and 2). **a**, **b** VEGF-C mediated migration (both siRNAs $P < 0.001$) ($n = 8$ per condition). Scale bar: 500 μm. **c** Tube formation: CD36siRNA1: branch points $P = 0.047$; tube length $P = 0.04$ versus Ctrl siRNA ($n = 8–6$ per condition). **d** Fully confluent Ctrl siRNA and CD36 siRNA 1 LECs ($n = 3$/condition) are serum starved (6 h) and exposed to VEGF-C (100 ng/ml) or VEGF-A (50 ng/ml) for 20 min. Immunohistochemistry of VE-cadherin depicts junctional morphology in Ctrl and CD36 siRNA LECs following treatments. **e** VE-cadherin and CD36 expression in Ctrl and CD36 siRNA LECs. **f** Western blot analysis and (g) densitometric quantification of CD36 ($P < 0.001$), VEGFR2, p-VEGFR2$^{Y1175}$ ($P < 0.05$), AKT, p-AKT$^{S473}$ ($P < 0.01$) expression in Ctrl and CD36 siRNA LECs ($n = 3$ per condition) following the addition of VEGF-C (100 ng/ml, 20 min). Actin protein is the loading control. VE-cad, VE-cadherin; kDa, kilodalton. Data are representative of two independent experiments and are means ± SE. Statistical significance is determined by two-tailed Student $t$ test. *$P < 0.05$; **$P < 0.01$; ***$P < 0.001$.

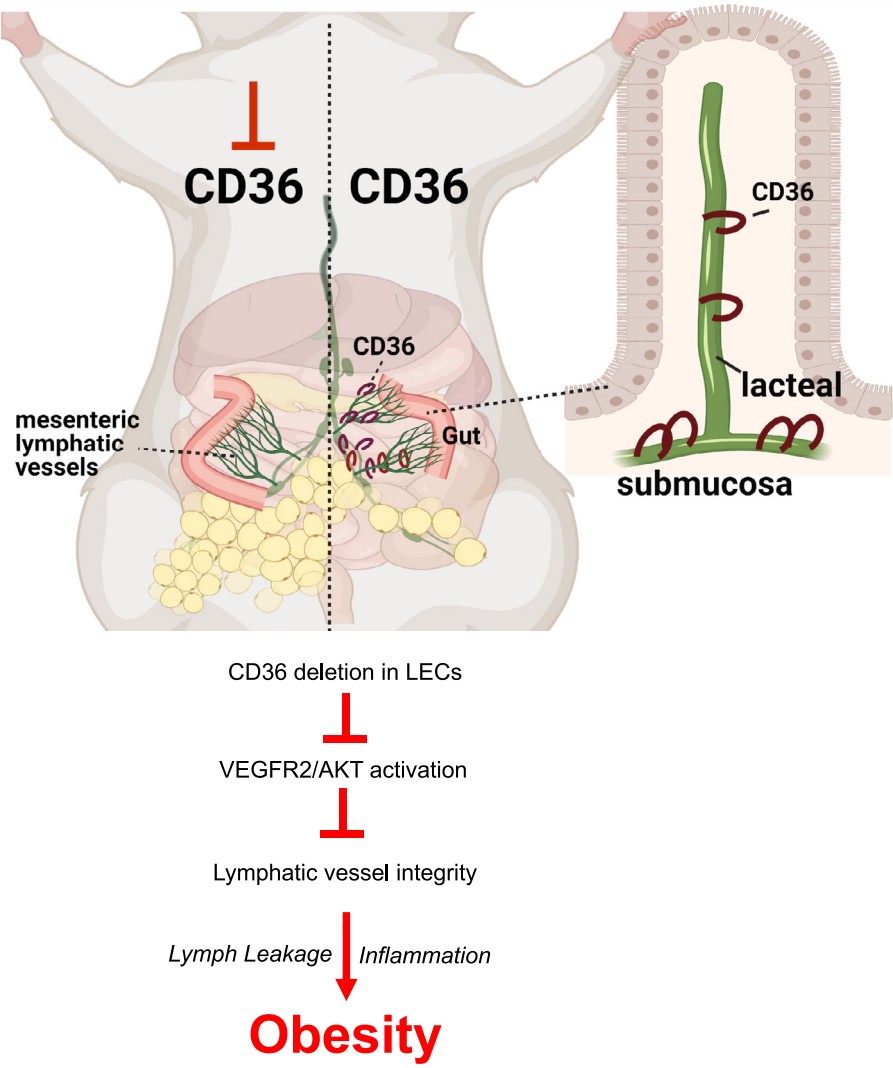

**Fig. 9 CD36 deletion in lymphatic endothelial cells associates with obesity and glucose intolerance.** CD36 expression in lymphatic endothelial cells (LECs) follows an increasing gradient from gut mucosa to submucosa to collecting mesentery vessels. CD36 deletion in LECs in adult mice ($Cd36^{\Delta LEC}$) increases discontinuity of VE-cadherin junctions and associates with lymph leakage in the mesentery. Inflammation of visceral fat, age-associated obesity and glucose intolerance are observed in 20-week-old $Cd36^{\Delta LEC}$ mice. The mechanism likely involves reduced signaling of VEGF-C in LECs, which leads to disorganization of LEC junctions and impairs cell migration and tube formation. Image created using BioRender.com.

protocols for the animal studies were reviewed and approved by the animal Ethical Committees at Washington University School of Medicine.

**Flow cytometry and LEC sorting**. Intestines were separated off the mesentery. The jejunum was then opened longitudinally and, after removal of Peyer's patches, gently scraped off to physically separate the mucosal layer (lacteals) from the submucosa by using a cell scraper. The submucosa was washed twice with DTT/EDTA on a shaker at 37 °C, followed by enzymatic digestion using the Miltenyi Lamina Propria Dissociation kit (Miltenyi Biotech) for 30 min, per manufacturer's instructions, whereas gut mucosa and mesentery were directly digested for 30 min. The three cell suspensions were stained with an antibody cocktail (Supplementary Table 1). CD36 expression was measured on LECs identified as CD45−CD90.2+CD31+ [70]. Doublets and dead cells, positive for LIVE/DEAD Fixable Aqua (Life Technologies), were excluded from analyses. Cells were analyzed using FACS Fortessa (BD Biosciences) and FlowJo v10 software (Treestar). LECs were sorted (BD FACSAria II, BD Biosciences) by the tdTomato-positive population in jejunum of control and $Cd36^{\Delta LEC}$ mice.

**Endothelial cell tube formation and migration assays**. Human Dermal LECs, from PromoCell were cultured in EBM-2MV medium (Lonza). For siRNA treatment, the LECs were grown to 60–80% confluence and transfected with either control or CD36 Silencer® Pre-designed siRNA, as per the manufacturer's instructions (Ambion). The Boyden chamber migration assay was performed using serum-starved Ctrl and CD36 siRNA treated LECs (50,000 cells/insert) grown on transwell inserts (8 μm pore size) coated with 10 μg/ml fibronectin (30 min at

37 °C). Migration was stimulated by adding 100 ng/ml VEGF-C (R&D System) and 25 ng/ml β-FGF to the lower well. LECs migrated for 24 h then supernatants were aspirated, the inserts washed and fixed with ice-cold methanol and LECs stained with Hoechst dye. Images of the lower side of inserts were taken and nuclei of migrated cells/well quantified using ImageJ v1.53i software (NIH). Tube formation was assayed in basal medium supplemented with 2% FCS, 100 ng/ml VEGF-C, and 25 ng/ml β-FGF. Ctrl and CD36 siRNA-treated LECs (25,000 cells/well) were seeded in 48-well plates with wells pre-coated with growth factor-reduced Matrigel (150 μl solidified at 37 °C for 30 min). Bright field images of capillary-like tubes were taken after 12 h at 37 °C for quantification. Mean tube length and branch points were quantified using ImageJ v1.53i software.

**Seahorse**. Oxygen consumption rate (OCR) and basal extracellular acidification rate (ECAR) values were obtained using the XFp Mito Stress Kit using the Seahorse XF96 Extracellular Flux Analyzer (Seahorse Biosciences). Briefly, LECs ($2.5 \times 10^5$ cells/well) were plated in XF96 cell plates 24 h before the assay and treated 16 h with/without 100 ng/ml VEGF-C (R&D System). Before assay, cells were switched to bicarbonate free media supplemented with pyruvate (1 mM), glucose (10 mM), and glutamine (2 mM) and kept 1 h in a non-CO$_2$ incubator at 37 °C. Oligomycin (3 μM), carbonyl cyanide-p trifluoromethoxyphenylhydrazone (FCCP, 2.5 μM) and antimycin A/rotenone (2 μM) were added to the appropriate injection ports.

**Western blotting**. Overnight serum-starved LECs were treated with 100 ng/ml VEGF-C (R&D System) for 20 min at 37 °C, lysed (20 min) in cold buffer (20 mM Tris-HCL, pH 7.5, 150 mM NaCL, 1% Triton X-100, 60 mM octyl β-D-

glucopyranoside, 200 μM sodium orthovanadate, 50 mM NaF, 1 mM PMSF, and 1 μg/ml protease inhibitor mix) and the cleared lysates ($10,000 \times g$, 10 min) assayed for protein (Pierce Biotech). Proteins separated on 4–20% gradient gels were transferred to polyvinylidene fluoride membranes (Merk Millipore), blocked (Li-COR Biosciences) 1 h at room temperature before adding primary antibodies (Supplementary Table 1) overnight at 4 °C. Infrared dye–labeled secondary antibodies were added for 1 h at room temperature. Protein signals were detected using the Li-Cor Odyssey Infrared (Li-COR Biosciences) and quantified using Image Studio Lite v5.2 Software.

**RNA extraction and qRT-PCR.** RNA was extracted using TRIzol (Invitrogen) and subjected to cDNA reverse transcription. Q-RT-PCR was performed using Power SYBR Green PCR Master Mix on a 7500 Fast Real-Time PCR System (Applied Biosystems). Relative mRNA fold changes were calculated using standard δCt. Primer sequences are listed in (Supplementary Table 2).

**Immunohistochemistry and whole-mount imaging.** Mice jejunums opened longitudinally, were fixed (4% formaldehyde) and paraffin embedded. Sections (5 μm) were deparaffinized and had antigen retrieval (99 °C, 18 min) in a pressurized chamber (Biocare Medical). Sections were incubated in donkey serum (2%) and BSA (3%) for 1 h at room temperature, and then overnight (4 °C) with primary antibodies (Supplementary Table 1) followed by fluorescently labeled (Alexa Fluor) secondary antibodies (1:250). For whole-mount staining, jejunums were fixed (4% formaldehyde for 2–4 h) and incubated in buffer with 5% non-immune donkey serum, 0.1% Triton-X, 1% BSA, and 0.05% $NaN_3$ in PBS (1 h, room temperature) then with primary antibodies in blocking buffer (2 days, 4 °C), washed with PBST (0.1% TritonX-100 in PBS, $3 \times 45$ min) and fluorescent secondary antibodies (Invitrogen) were added overnight at 4 °C. Following PBST washes, tissues were post-fixed with 1% formaldehyde (10 min, room temperature) and PBS washed. As primary antibody control, staining employing each fluorescent secondary Abs was used. For LEC VE-cadherin staining, cells were cultured in complete media and used at 100% confluency. After serum starvation for 6 h, LECs were treated with VEGF-C (100 ng/mL), or VEGF-A (50 ng/mL) (both from R&D System) for 20 min. Cells were then fixed (4% formaldehyde, 10 min, room temperature), permeabilized (0.1% Triton X-100/PBS, 2 min), and blocked (1% BSA/PBS, 30 min). Anti-VE-cadherin antibody (1 h) and fluorescent secondary antibody (30 min) were added at room temperature, each followed by $3 \times 10$ min PBS washes. Stained tissues and cells were mounted using the fluorescent mounting medium (DAKO Inc.) and whole-plane z-stack images acquired on a Zeiss LSM 880 Airyscan Confocal Microscope using ×20, ×40 or ×63 objectives. For lacteal structure, villi and lacteal length were measured with ImageJ v1.53i software (NIH).

**Analysis of lymphatic vessel function.** To analyze lymph transport by mesenteric lymphatic vessels, mice received intragastrically 0.1 mg of BODIPY FL $C_{16}$ (4,4-difluoro-5,7-dimethyl-4-bora-3a, 4a-diaza-s-indacene-3-hexadecanoic acid, Molecular Probes, Inc.) in heavy cream. Mice were sacrificed 2 h later and transport of fluorescent lipid visualized by Zeiss Stereo Lumar v.12 fluorescence stereomicroscopy.

**Statistical analyses.** All data shown are means ± standard error (SE). Statistical significance was evaluated using two-sided Student's $t$ test with a $p$ value of ≤0.05 indicating significant differences. Graphpad Prism software (version 8.4.3) was used for all statistical analyses and to plot quantitative data.

**Reporting summary.** Further information on research design is available in the Nature Research Reporting Summary linked to this article.

## Data availability

Source data files are provided with this paper and all relevant data can be obtained from the corresponding authors Nada Abumrad or Vincenza Cifarelli upon reasonable request. Source data are provided with this paper.

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

## Acknowledgements

This work was supported by National Institutes of Health grants DK060022 (NAA), DK111175 (NAA) and by the Leducq Foundation Transatlantic Network of Excellence "Lymph vessels in obesity and cardiovascular disease" (HGA). We acknowledge Pilot and Feasibility grants from the Nutrition and Obesity Research Center (NORC) P30 DK056341 (VC) and the Digestive Diseases Research Cores Center P30 DK052574 (VC) at Washington University, and the assistance of Washington University NORC Cellular and Molecular Biology Core and Animal Model Research Core. R.S.C. was supported by a fellowship award "FA-2020-01-IBD-1 from the Lawrence C. Pakula, MD IBD Education & Innovation Fund". Confocal data were obtained in the Washington University Center for Cellular Imaging (WUCCI) using a Zeiss LSM 880 Airyscan confocal Microscope purchased with support from the Office of Research Infrastructure Programs (ORIP), NIH Office of the Director grant OD021629.

## Author contributions

V.C., N.A.A., S.A.B., and H.G.A. designed the research studies and analyzed the data; V.C. conducted the experiments and analyzed the data; S.A.B. contributed experiments and data analysis; V.P., A.K., T.S., R.N., S.I., K.M.P., C.W.W., and R.C. contributed to data collection; M.C. and G.J.R. contributed to project design; V.C. and N.A.A. wrote the manuscript. All authors reviewed the manuscript.

## Competing interests

The authors declare no competing interests.
