## [Peer Review File · Nature Communications]

REVIEWER COMMENTS

Reviewer #1 (Remarks to the Author):

Cifarelli et al. report functions of the scavenger receptor CD36 in lymphangiogenesis. Using global CD36 ko mice and Prox1-CreERT2 driven conditional CD36 deletion, they demonstrate that lymphatic capillaries in the small intestine (lacteals) and mesenteric lymphatic collectors develop abnormally, leading to deficient chylomicron transport and lymph leakage, thereby inducing visceral adipose tissue accumulation, inflammation and obesity. The study is well written and interesting, but a number of concerns must be addressed before publication can be recommended.

Specific comments:

1. Fig. 1: CD36 expression in lymphatic endothelial cells (LECs) is well illustrated in Fig 1a, but not all LYVE1+ cells appear to stain positive for CD36 (see top right corner in Fig1a right panel). The FACS analysis appears to support a heterogeneous CD36 expression in LECs, but this is not well explained. Can the authors expand on the illustration of CD36 LEC expression, by including more images and knockout (global and Prox1-Cre-induced) validations to demonstrate antibody specificity. Specifically, it would be interesting to know if CD36 is expressed differentially between lymphatic capillaries and collecting vessels? Are lymphatic valves CD36 positive? Is staining restricted to intestinal lymphatics? Is CD36 expression in LECs developmentally regulated? The legend for panel b is hard to understand, please provide better explanation for the FACS analysis. Panels c and d: what age are the mice? Were littermate controls used here?
2. The phenotypic differences between global and Prox1-Cre induced CD36 mutants deserve more consideration. First, global CD36 ko mice show shorter lacteals, but the Prox1-Cre induced ones do not have apparent lacteal length reduction. Since lacteals are known to proliferate continuously during adult life, and the authors claim that CD36 knockdown in vitro reduces LEC proliferation, one would expect a lacteal length reduction even in adult-stage CD36 deletions. Second, since Prox1-Cre-driven CD36 deletion leads to leak from mesenteric collectors, one wonders if this also occurs in the global CD36 ko mice. More carefully comparing the different mutant strains may help better understanding lymphatic CD36 function.
3. Prox1 is expressed in various other cells besides lymphatic endothelial cells, including hepatic and neuronal cell populations that may affect metabolism. The authors should discuss if CD36 deletion in these cells could contribute to the metabolic phenotypes observed, and avoid saying 'lymphatic-specific' deletion when using Prox1-Cre drivers.
4. The author state in their methods that they use Cre+ flox- mice as controls. It is not possible to get flox- and flox+/+ litter-mates, unless the parents were both flox+/- . What was the breeding strategy to obtain the mice, and why didn't the authors use Cre-/flox+/+ mice?
5. Fig 2: it seems that the VE-Cadherin staining in the CD36 mutants is highly fragmented and overall reduced compared to controls. I would not classify the mutant junctions as "buttons". In fact, it seems impossible to trace the junction length from this picture, because VE-Cadherin staining is not linear at all. The same is true for the conditional KO (Fig 4a).
5. Why do the collectors leak? What is the morphology of VE-Cadherin in the collecting lymphatics? Is the junction morphology affected? Is the SMA coverage defective? Are the valves intact in the mutant mice? In Figure 4d, is the leakage homogenous throughout the collectors, or does it accumulate centrally?
6. The in vitro experiments suggest that LECs deficient for CD36 are generally dysfunctional (in terms of proliferation and migration). Adding a proliferation test in vitro and measuring proliferation in the mouse mutants in vivo would be useful. They also show oxidative metabolism defects, which however are not linked to the in vivo phenotype. My impression from Fig 4 is that functionally the collectors leak, which leads to secondary IP inflammation and obesity; however, it is unclear how this finding relates to any oxidative metabolism defects in vivo.
7. Fig 6 has a number of issues. First, the junctions in the siCD36 cells look thinner, supporting the idea that the si cells have lower levels of VE-Cadherin at their junctions. Second the lack of transformation of junctions to zippers in the si cells is not convincing, as a tiny fraction of the total junction length is "button like" and the junctions look mostly similar between the si and the controls. There is no information as to how confluent the cells are. The "untreated" condition is in fact treated with complete media, which I assume contains VEGFC. The authors should show the junctions of serum-starved cells as a control, and they should also try to stimulate with VEGF,

which is a stronger transducer of LEC junctions towards zippers.
8. Figure S2: is the diameter of the KO lacteals larger?

Reviewer #2 (Remarks to the Author):

The manuscript by Cifarelli et al. reports the role of CD36 in lymphatic endothelial cell integrity in mice. The authors showed that LEC-selective deletion of CD36 by a Prox1-CreER driver disrupts VE-cadherin junctions in LEC, and it increases the permeability of lymphatic vessels as well as adipose tissue expansion. CD36 deletion in LECs also decreases proliferation and mitochondrial FA oxidation. Overall, the data are convincing and well support the conclusion of the paper. On the other hand, several issues are identified – the authors wish to address the following point to strengthen the paper.

1. The result that CD36 deletion by Prox1-Cre causes visceral WAT expansion is intriguing, while it is unclear if the change in WAT mass is selective to visceral WAT or all adipose tissues, such as subcutaneous adipose tissues and brown adipose tissues. If selective to visceral WAT, the authors should discuss how TG goes to visceral WAT preferentially.
2. Relating to the above question, it is unclear if the obese phenotype is associated with any changes in energy expenditure or food intake. The authors should examine the metabolic rate of KO mice.
3. It is possible that increased inflammation and impaired glucose intolerance in KO mice is merely a consequence of obesity. The authors should examine if these phenotypes are seen before the KO mice develop an obese phenotype. This is important to exclude the possibility that impaired glucose tolerance is driven by accumulated lipids in the adipose tissues, rather than abnormalities in other tissues/cell types besides LEC.
4. An important data would be to examine how CD36 expression in the LEC is regulated under an obese condition. This will provide a pathophysiological context in which the CD36 pathway in LEC is involved in the pathogenesis of adipose tissue expansion and obesity.

Reviewer #3 (Remarks to the Author):

This study documents – for the first time – the expression of CD36 also in intestinal lymphatic endothelial cells. Furthermore, it is described that mice with a specific deletion of CD36 in this cell type suffer from decreased lymphatic integrity, display delayed and prolonged intestinal lipid absorption, and finally were more prone to develop obesity and type 2 diabetes.

These novel observations are of interest, but also evoke a number of comments.

1. CD36 is found in LEC indeed. Have the authors confirmed its presence at the plasma membrane of these cells? In other cell types it has been described that CD36 recycles between a subcellular compartment (endosomes) and the plasma membrane so as to regulate its functional presence at the plasma membrane. Has such mechanism also been found in LEC?
2. For other tissues, e.g. muscle, it has been reported that deletion of CD36 was accompanied by the compensatory upregulation of other fatty acid transporters, such as FATPs (6 distinct types exist). Did the authors evaluate such compensatory changes in LEC?
3. A main question remains whether the reported effects in the CD36-deltaLEC mice are related to CD36 directly, i.e., its putative function in intestinal lipid transport, or indirectly, due to impaired energy provision of the LEC. In case of the latter, because of fatty acid substrate deficiency and a resulting energy deficit, LEC may not function appropriately leading to loss of cell integrity and perhaps lymphatic integrity. However, can't LEC use alternative substrates for energy provision instead? Has this been explored?

4. In CD36-delta LEC mice lipid absorption is delayed and prolonged. This would be expected to allow more time for peripheral tissues (including adipose tissue) to take up and store and/or oxidize the lipids. Still, this prolonged duration is associated with increased obesity and insulin resistance rather than a decrease which seems counter intuitive. How can this be explained?

5. Marked differences are known to exist between females and males regarding visceral and subcutaneous fat distribution. Interestingly, both female and male mice were used (line 310). However, no report was made whether gender differences were observed, or not. This aspect should be considered in the present work.

5. A graphical abstract or summary figure would help the reader to get the main point of the new work.

6. Typo online 168: "nest" should read "next".

Reviewer 1:

We thank the reviewer for all comments and suggestions, which helped improve the manuscript. All Changes in the revised manuscript are highlighted in red font.

Specific Comments:

1- Fig.1: CD36 expression in lymphatic endothelial cells (LECs) is well illustrated in Fig 1a, but not all LYVE1+ cells appear to stain positive for CD36 (see top right corner in Fig1a right panel). The FACS analysis appears to support a heterogeneous CD36 expression in LECs, but this is not well explained.

We agree that CD36 has heterogeneous expression in intestinal LECs. This is in line with heterogeneity of gene expression and function of lymphatic ECs. The heterogeneity applies to expression of metabolic genes and to individual lymphatic markers (Kalucka J, Cell 2020, PMID: 32059779). This is included in the revised manuscript on page 5, line 2-4 and on page 12, lines 6-7.

- The legend for panel b is hard to understand, please provide better explanation for the FACS analysis. Panels c and d: what age are the mice? Were littermate controls used here?

Legend of Figure 1 now includes explanation for the FACS analysis in LECs isolated from gut mucosa and submucosa, and mesentery vessels (Fig. 1b-c). Age of the mice used in experiments of Fig. 1a-c is now included in the legend. Sex- and age-matched C57BL/6 wildtype (WT) mice and *Cd36*^{-/-} mice are bred and maintained in the same facility and details have been added to Materials and Methods, on page 16, lines 3-5.

- Can the authors expand on the illustration of CD36 LEC expression, by including more images and knockout (global and Prox1-Cre-induced). Validations to demonstrate antibody specificity.

We added an additional image with CD36/Lyve-1 immunostaining in the submucosa, Fig. 1a (new bottom panel). We used the APC anti-mouse CD36 antibody (clone HM36, Biolegend cat.# 102812) for flow cytometry as reported by others (De Silva N, 2016, PNAS, PMID: 27457956; Misumi I, 2019, Cell Rep, PMID: 30970254). We tested the CD36 Ab in *Cd36 null (Cd36*^{-/-}) LECs and observed no signal (new Fig.1c). We also added to our analysis an isotype control (Armenian Hamster IgG Isotype Ctrl, Biolegend cat # 400911) at the same concentration of the CD36 Ab, (new Fig. 1c). This is now reported in Fig.1 legend and in the revised manuscript, on page 5, lines 8-9.

- Are lymphatic valves CD36 positive? Is staining restricted to intestinal lymphatics? Is CD36 expression in LECs developmentally regulated?

We cannot detect CD36 expression in lymphatic valves. Staining for FoxC2, which identifies valves, did not show changes in valve morphology between WT and *Cd36*^{-/-} collecting lymphatics (new Supplementary Fig. 2).

We also performed flow cytometric analysis to compare CD36 expression in mucosa (lacteals) and submucosa lymphatics and in collecting lymphatics of mesentery (new Fig. 1b-c). CD36 expression in mesentery LECs is 3-fold higher (MFI: 9010) than in the mucosa (MFI: 2952) and 2-fold higher than in the submucosa (MFI: 4027). This is discussed in the revised manuscript on page 5, lines 4-8.

We are not aware of studies that tested whether CD36 expression in lymphatic vessels is regulated during development. This is not addressed in the present investigation, which is focused on role of CD36 expression in lymphatic function. Based on current findings, we agree that this would be worth exploring in the future.

2- The phenotypic differences between global and Prox1-Cre induced CD36 mutants deserve more consideration.

- First, global CD36 ko mice show shorter lacteals, but the Prox1-Cre induced ones do not have apparent lacteal length reduction. Since lacteals are known to proliferate continuously during adult life, and the authors claim that CD36 knockdown in vitro reduces LEC proliferation, one would expect a lacteal length reduction even in adult-stage CD36 deletions.

We tested whether CD36 deletion affects LEC proliferation *in vivo* by measuring BrdU incorporation in gut lymphatics of WT and *Cd36*^{-/-} mice. Proliferation did not differ between mouse groups *in vivo* and similar results were obtained in cultured LECs *in vitro*. This may not be surprising based on our new data showing that basal

glycolytic flux is significantly increased in CD36 siRNA LECs where fatty acid oxidation (FAO) is suppressed (new Fig. 7f-g). Similarly, gut LEC isolated from *Cd36^{AL}* mice have increased expression of the glucose transporter *Glut1* gene and decreased expression of FAO genes, as compared to control LECs (Fig. 7j). The LECs have high glycolytic rates that greatly exceed those of FAO and primarily fuel proliferation (De Bock K, Cell 2013, PMID: 23911327). Thus, it is unlikely that the shorter lacteals in *Cd36^{-/-}* mice can be explained by a defect in proliferation. It is possible that differences in gradient of angio- and lymphangiogenic growth factors, important for LEC migration and tube formation, caused the shorter lacteals in *Cd36^{-/-}* mice. We show that VEGF-C stimulation of tube formation and migration is blunted when CD36 is silenced *in vitro*. Based on these new data, we revised our conclusion on CD36 regulation of LEC proliferation and removed the MTT assay as this assay measures proliferation indirectly. This is now reported in the revised manuscript on page 12, lines 12-19.

- Second, since Prox1-Cre-driven CD36 deletion leads to leak from mesenteric collectors, one wonders if this also occurs in the global CD36 ko mice. More carefully comparing the different mutant strains may help better understanding lymphatic CD36 function.

We investigated presence of lymphatic vessel leakage in *Cd36^{-/-}* mice by microscopy after intragastric load of the fluorescently-labelled BODIPY C16 fatty acid. We observed little to no fluorescence in mesentery lymphatics by microscopy (data not shown). Blood sampling over time showed faster appearance of fluorescence in the circulation as compared to WT mice (new Supplementary Fig. 1b). These data are consistent with our earlier findings showing that when *Cd36^{-/-}* mice are infused with intra-duodenal lipid, much less lipid reaches the cannulated mesenteric lymph duct, as compared with WT mice. In addition, *Cd36^{-/-}* mice display abnormally fast appearance of absorbed dietary triglycerides in the circulation (Drover VA et al. JCI 2005, PMID:15841205 and Nauli A et al., Gastroenterology 2006, PMID: 17030189). We believe that in *Cd36^{-/-}* mice the absorbed lipids reach the circulation mainly through the portal blood and independently of lymphatic transport. This could be facilitated by the compromised integrity of blood endothelial vessels lacking CD36 (Tie2 driven EC deletion) (Cifarelli V et al. CMGH 2017, PMID:28066800). In this study, we show that lacteal junctions in *Cd36^{-/-}* mice are highly fragmented (Fig. 2d) which is likely to impede their function in lipid uptake and explains previous findings. By comparison, the *Cd36^{AL}* mice, have lacteal LEC junctions that are more discontinuous than in control mice, but not as extensively fragmented as in *Cd36^{-/-}* mice. Lymph transport of lipid occurs in these mice but is defective. We now compare the two mouse models and discuss the several factors underlying the differences in lymphatic phenotypes. This is discussed in the revised manuscript on page 13 line 25, and page 14, lines 1-9.

3- Prox1 is expressed in various other cells besides lymphatic endothelial cells, including hepatic and neuronal cell populations that may affect metabolism. The authors should discuss if CD36 deletion in these cells could contribute to the metabolic phenotypes observed, and avoid saying ‘lymphatic-specific’ deletion when using Prox1-Cre drivers.

We have added discussion related to expression of Prox1 in cells other than LECs on page 15 lines 8-10. As suggested, we now refer to CD36 “deletion in LEC” without “specific”.

4- The authors state in their methods that they use Cre+ flox- mice as controls. It is not possible to get flox- and flox+/+ littermates, unless the parents were both flox+/- . What was the breeding strategy to obtain the mice, and why didn't the authors use Cre-/flox+/+ mice?

We crossed Cre+ and flox+/- mice to obtain Cre+/flox+/, and Cre+/flox-/- (controls) from the same litter. We did not use Cre-/flox+/+ mice as Cre activation is needed for tdTomato expression to label lymphatics.

5- Fig 2: it seems that the VE-Cadherin staining in the CD36 mutants is highly fragmented and overall reduced compared to controls. I would not classify the mutant junctions as "buttons". In fact, it seems impossible to trace the junction length from this picture, because VE-Cadherin staining is not linear at all. The same is true for the conditional KO (Fig 4a).

As suggested, we now describe LEC junctions as discontinuous or fragmented, throughout the revised manuscript on page 5 lines 21-23, and page 8 lines 10-13.

6- What is the morphology of VE-Cadherin in the collecting lymphatics? Is the junction morphology affected? Is the SMA coverage defective? Are the valves intact in the mutant mice? In Figure 4d, is the leakage homogenous throughout the collectors, or does it accumulate centrally?

We examined VE-Cadherin morphology in collecting vessels. We found that as with lacteals, CD36 deletion associates with more fragmented junctional VE-cadherin in collecting lymphatics of *Cd36*^{-/-} and *Cd36*^{ΔLEC} mice versus respective controls. These data are included in new Figs 2d and 4b, and in the revised manuscript, on page 5 lines 21-23, and page 8 lines 10-13.

With respect to smooth muscle actin- α and valve morphology (staining of FoxC2) in mesenteric lymphatic vessels, both were similar in WT and *Cd36*^{-/-} mice and were not investigated further in *Cd36*^{ΔLEC} mice. These data are in new Supplementary Fig. 2, and on page 6, lines 4-6. Lymph leakage was observed in segments of mesenteric vessels and central accumulation of lymph was not observed.

7- The *in vitro* experiments suggest that LECs deficient for CD36 are generally dysfunctional (in terms of proliferation and migration). Adding a proliferation test *in vitro* and measuring proliferation in the mouse mutants *in vivo* would be useful. They also show oxidative metabolism defects, which however are not linked to the *in vivo* phenotype. My impression from Fig 4 is that functionally the collectors leak, which leads to secondary IP inflammation and obesity; however, it is unclear how this finding relates to any oxidative metabolism defects *in vivo*.

Whether CD36 deletion in LECs affects cell proliferation *in vivo* has been addressed in comment 2.

To address whether the *in vitro* observed defects in oxidative metabolism also apply to *in vivo*, gut LECs were sorted from 20-week-old *Cd36*^{ΔLEC} mice and controls for gene expression analysis. *Cd36*^{ΔLEC} LECs have decreased expression of key genes of fatty acid oxidation, *Cpt1a*, *Acs1l* and *Vlcad*, while gene expression of *Glut1* was increased, as compared to LECs from control mice. CD36 requirement for upregulation of FAO has been reported in several cell types and tissues (Ibrahimi, A. J Biol Chem 274, 26761-26766 (1999); Campbell, S.E. J Biol Chem 279, 36235-36241 (2004); Liu, T.F., J Biol Chem 287, 25758-25769 (2012); Huang, S.C., et al. Nat Immunol 15, 846-855 (2014). The new data are in Fig. 7j and in the revised manuscript on page 11 lines 4-6.

We agree that leak in the lymphatic collectors might have promoted inflammation and then obesity. We conducted additional experiments to compare glucose disposal and adipose tissue gene expression of key markers of inflammation and fibrosis in 11- and 20-week-old *Cd36*^{ΔLEC} mice. At 11 weeks, when body weight did not differ, glucose disposal was similar for *Cd36*^{ΔLEC} mice and controls (new Fig. 3e). However, adipose gene expression of tumor necrosis factor- α (*Tnf*) and of the macrophage marker *Emr1* (EGF module-containing mucin-like receptor 1) was already increased in 11-week-old *Cd36*^{ΔLEC} mice while expression of lipid storage genes (*LpL* and *PPAR γ*) did not change as compared with age-matched controls (Fig. 3g). This suggests that adipose tissue inflammation precedes the onset of obesity. These new data (Fig. 3e and 3g) are discussed on page 7, lines 11-21 and from page 14 line 23-25 to page 15 lines 1-2.

8- Fig 6 has a number of issues. First, the junctions in the siCD36 cells look thinner, supporting the idea that the si cells have lower levels of VE-Cadherin at their junctions. Second the lack of transformation of junctions to zippers in the si cells is not convincing, as a tiny fraction of the total junction length is "button like" and the junctions look mostly similar between the si and the controls. There is no information as to how confluent the cells are. The "untreated" condition is in fact treated with complete media, which I assume contains VEGFC. The authors should show the junctions of serum-starved cells as a control, and they should also try to stimulate with VEGF, which is a stronger transducer of LEC junctions towards zippers.

We appreciate the reviewer's suggestions. Fig. 6 is now Fig. 8. VE-cadherin protein expression by Western blot did not differ in Control and CD36 siRNA LECs as shown in new Fig. 8f and discussed in revised manuscript on page 11, lines 10-15. As suggested, we serum-starved the cells for 6 h and included VEGF-A treatment in addition to VEGF-C. VE-cadherin staining is conducted in 100% confluent LECs. This is reported in Methods, pages 19-20, lines 23 and in revised legend of Fig. 8.

9- Figure S2: is the diameter of the KO lacteals larger?

We measured lacteal width in *Cd36*^{ΔLEC} and controls mice and found that lacteal diameter trended higher in *Cd36*^{ΔLEC} mice as compared with controls but did not reach significance.

Reviewer #2 (Remarks to the Author):

We thank the reviewer for all comments and suggestions, which helped improve the manuscript. All Changes in the revised manuscript are highlighted in red font.

The manuscript by Cifarelli et al. reports the role of CD36 in lymphatic endothelial cell integrity in mice. The authors showed that LEC-selective deletion of CD36 by a Prox1-CreER driver disrupts VE-cadherin junctions in LEC, and it increases the permeability of lymphatic vessels as well as adipose tissue expansion. CD36 deletion in LECs also decreases proliferation and mitochondrial FA oxidation. Overall, the data are convincing and well support the conclusion of the paper. On the other hand, several issues are identified – the authors wish to address the following point to strengthen the paper.

1- The result that CD36 deletion by Prox1-Cre causes visceral WAT expansion is intriguing, while it is unclear if the change in WAT mass is selective to visceral WAT or all adipose tissues, such as subcutaneous adipose tissues and brown adipose tissues. If selective to visceral WAT, the authors should discuss how TG goes to visceral WAT preferentially.

We observed relatively more expansion of visceral adipose tissue but subcutaneous depots also expanded in *Cd36^{AL}* mice as compared to controls. Images and quantification related to subcutaneous adipose tissue are now included in new Fig. 3c-d and discussed in the revised manuscript on page 7, lines 9-10.

- Relating to the above question, it is unclear if the obese phenotype is associated with any changes in energy expenditure or food intake. The authors should examine the metabolic rate of KO mice.

We investigated if the obese phenotype in *Cd36^{AL}* mice associates with changes in food intake and energy expenditure by placing 11- and 20-week-old *Cd36^{AL}* mice and their littermate controls in metabolic cages for 48 h. The 11-week-old *Cd36^{AL}* mice compared to controls, showed reduced energy expenditure (light period) (new Fig. 4a, b, c) with differences becoming more significant at 20-week of age (new Fig. 4d-f). Food intake and activity were similar (data not shown). Thus, a small reduction in energy expenditure occurs after induction of CD36 deletion in *Cd36^{AL}* mice and might contribute to the weight increase. These new data (new Fig. 4a-f) are discussed in the revised manuscript on page 7 line 23-24, page 8 lines 1-5 and page 15, lines 5-7.

2- It is possible that increased inflammation and impaired glucose intolerance in KO mice is merely a consequence of obesity. The authors should examine if these phenotypes are seen before the KO mice develop an obese phenotype. This is important to exclude the possibility that impaired glucose tolerance is driven by accumulated lipids in the adipose tissues, rather than abnormalities in other tissues/cell types besides LEC.

We examined glucose disposal and adipose tissue gene expression (lipid storage and inflammation) at 1 and 10 weeks after induction of *Cd36* deletion, in 11- and 20-week-old *Cd36^{AL}* mice, respectively. At 11 weeks, there was no difference in glucose disposal between *Cd36^{AL}* mice and controls (new Fig. 3e). However, expression of *Tnf* (TNF-alpha) and of the macrophage marker *Emr1* was already increased in visceral fat of *Cd36^{AL}* mice while expression of lipid storage genes (*Pparg* and *Lpl*) was not (new Fig. 3g). Thus, in *Cd36^{AL}* mice adipose tissue inflammation precedes obesity. The new data are discussed in the revised manuscript on page 7, lines 11-21 and from page 14 line 23-25 to page 15 lines 1-2.

3- An important data would be to examine how CD36 expression in the LEC is regulated under an obese condition. This will provide a pathophysiological context in which the CD36 pathway in LEC is involved in the pathogenesis of adipose tissue expansion and obesity.

Effects of 12-week high fat diet (HFD) on CD36 expression, body weight, epididymal fat mass and inflammation were examined in WT, *Cd36^{-/-}*, *Cd36^{AL}* and control mice. The HFD did not increase CD36 protein level in mucosa LECs (lacteal) but CD36 level was upregulated 2.1 and 2.5-fold in LECs from submucosa and mesenteric lymphatics, compared to mice fed chow (Fig. 6a). The HFD increased body weight and epididymal fat mass in WT and controls, and these increases were exceeded in *Cd36^{AL}* mice. The *Cd36^{-/-}* mice were protected (new Fig. 6b-c) as before (Hajri, *T Diabetes*, 2007, PMID: 17440173). Expression of proinflammatory genes in epididymal fat increased in HFD fed WT and controls and further upregulation of all genes occurred in *Cd36^{AL}* mice. *Cd36^{-/-}* mice did not accumulate epididymal fat and displayed low tissue inflammation (new Fig. 6d-e). Thus, HFD increases LEC CD36 in collecting lymphatics, which optimizes transport of absorbed fat but likely contributes to

weight gain. Data are now included in new Fig. 6a-e, and in revised manuscript pages 9, lines 12- 24 and 10 lines 1-2.

Reviewer #3 (Remarks to the Author):

We thank the reviewer for all comments and suggestions, which helped improve the manuscript. All Changes in the revised manuscript are highlighted in red font.

This study documents – for the first time – the expression of CD36 also in intestinal lymphatic endothelial cells. Furthermore, it is described that mice with a specific deletion of CD36 in this cell type suffer from decreased lymphatic integrity, display delayed and prolonged intestinal lipid absorption, and finally were more prone to develop obesity and type 2 diabetes. These novel observations are of interest, but also evoke a number of comments.

1- CD36 is found in LEC indeed. Have the authors confirmed its presence at the plasma membrane of these cells? In other cell types it has been described that CD36 recycles between a subcellular compartment (endosomes) and the plasma membrane so as to regulate its functional presence at the plasma membrane. Has such mechanism also been found in LEC?

CD36 localization at the plasma membrane in LECs is confirmed by our flow cytometry studies which used non permeabilized cells. We also examined whether acute (20 min) VEGF-C treatment or 6 h serum starvation changes CD36 surface expression in LECs. Following VEGF-C treatment, flow-cytometric analysis showed that neither short-term serum starvation (6-h) or acute VEGF-C treatment (20 min) altered CD36 surface expression when compared to CD36 expression on LECs grown in complete media. These data are in Supplementary Figure 5 and in the revised manuscript on page 11, lines 20-22.

Additional studies that address comprehensively the potential regulation of CD36 internalization and cellular trafficking in LECs are currently on-going in the laboratory.

2- For other tissues, e.g. muscle, it has been reported that deletion of CD36 was accompanied by the compensatory upregulation of other fatty acid transporters, such as FATPs (6 distinct types exist). Did the authors evaluate such compensatory changes in LEC?

We evaluated the expression of fatty acid transporters (FATPs/SLC27A1-6) in LECs following CD36 silencing. We found no change in gene expression of *Slca27a1*, *Slca27a3* and *Slca27a4* between Ctrl and CD36 siRNA HDLECs. Expression of *Slca27a5* and *Slca27a6* was not detectable in HDLECs as previously reported (Kazantzis M, BBA. 2012; PMID: 21979150). Data are now included in Fig. 7c and in the revised manuscript on page 10, lines 8-11.

3- A main question remains whether the reported effects in the CD36-deltaLEC mice are related to CD36 directly, i.e., its putative function in intestinal lipid transport, or indirectly, due to impaired energy provision of the LEC. In case of the latter, because of fatty acid substrate deficiency and a resulting energy deficit, LEC may not function appropriately leading to loss of cell integrity and perhaps lymphatic integrity. However, can't LEC use alternative substrates for energy provision instead? Has this been explored?

LECs can use both glycolysis and fatty acid oxidation (FAO) and both are important for LEC maintenance. Glycolytic rates are much higher than FAO rates (De Bock K, Cell 2013PMID: 23911327) and fuel energy production during proliferation and migration while FAO is used for nucleotide synthesis and histone acetylation during differentiation to promote transcription of key lymphatic genes (Wong BW Nature 201; PMID: 28024299). We showed that oxidative metabolism is reduced after CD36 knockdown in LECs. In response to the reviewer's comment, we examined basal glycolytic flux in these cells and its response to VEGF-C. Basal glycolysis was significantly increased in CD36 siRNA LECs as compared to Control siRNA LECs. However, unlike control cells, CD36 siRNA LECs did not increase glycolytic flux further in response to VEGF-C. These findings are supported by increased gene expression of glucose transporter 1 (6-fold) and key glycolytic enzymes hexokinase 2 (*HK2*, 1.8-fold), and aldolase A (*ALDOA*, 3-fold) in LECs after CD36 silencing. In addition, gene expression of these enzymes were increased by VEGF-C in Ctrl LECs but not in CD36 depleted LECs. These new data are in new Fig. 7f-i and discussed in the revised manuscript on pages 10, lines 17-25, and 11, lines 1-7.

4- In CD36-delta LEC mice lipid absorption is delayed and prolonged. This would be expected to allow more time for peripheral tissues (including adipose tissue) to take up and store and/or oxidize the lipids. Still, this prolonged duration is associated with increased obesity and insulin resistance rather than a decrease which seems counter intuitive. How can this be explained?

We conducted additional studies to clarify the obese phenotype in *Cd36*^{ALEC} mice. Disruption of lymphatic vessel integrity such as that we show after LEC CD36 deletion leads to subtle leakage of lymph (and immune cells) in the mesentery, promoting ectopic adipogenesis and spontaneous obesity (Harvey et al., 2005; Escobedo et al., 2016). We now show that deletion of CD36 in LECs induces upregulation of key pro-inflammatory genes (*Tnf* and *Emr1*) in adipose tissue already at 1 week after TAM treatment to induce CD36 deletion (11-week-old *Cd36*^{ALEC} mice) (new Fig. 3g) without impacting glucose disposal (new Fig. 3e). We believe that the proinflammatory changes are likely consequent to the leaked lymph diverting to adipose tissue immune cells and pathogens that would have normally trafficked to lymph nodes. These data suggest that expansion of visceral fat mass in *Cd36*^{ALEC} mice associates early with inflammation and macrophage infiltration, as increases in *Tnf* and *Emr1* expression precede increases in lipid storage genes and onset of obesity and glucose intolerance. The new data are discussed in the revised manuscript on page 7, lines 11-21 and from page 14 line 23-25 to page 15 lines 1-2.

We observed a small reduction in energy expenditure (light period) in 11-week-old *Cd36*^{ALEC} mice and the difference became more significant at 20-week of age (new Fig. 4). Food intake and activity between the two groups were similar (data not shown). The small reduction in energy expenditure in *Cd36*^{ALEC} mice could have contributed to the weight increase. These new data (new Fig. 4a-f) are discussed in revised manuscript on page 7 line 23-24, page 8 lines 1-5 and page 15, lines 5-7.

5- Marked differences are known to exist between females and males regarding visceral and subcutaneous fat distribution. Interestingly, both female and male mice were used (line 310). However, no report was made whether gender differences were observed, or not. This aspect should be considered in the present work.

We measured body weight, and body composition by DEXA in both female and male *Cd36*^{ALEC} mice and their respective controls. Body weight and fat mass were both increased in male *Cd36*^{ALEC} mice as compared with controls. Although to a less degree, similar trends were observed in female *Cd36*^{ALEC} mice as compared with controls. No changes in lean mass were observed in male or female *Cd36*^{ALEC} mice as compared with their respective controls. The data are now reported separated by mouse sex in Figure 3a-b and in revised manuscript on page 7, lines 4-9.

6- A graphical abstract or summary figure would help the reader to get the main point of the new work.

A graphical abstract is now included in Figure 9.

7- Typo on line 168: “nest” should read “next”.

This was corrected.

REVIEWERS' COMMENTS

Reviewer #1 (Remarks to the Author):

The authors have addressed my comments and I am pleased to recommend publication

Reviewer #2 (Remarks to the Author):

The authors provided new and convincing data that addressed all the reviewer's comments. No more comment from this reviewer.